# A functional model of adult dentate gyrus neurogenesis

Olivia Gozel[1,2,3†]*, Wulfram Gerstner[1]

[1]School of Life Sciences and School of Computer and Communication Sciences, Ecole Polytechnique Fédérale de Lausanne, Lausanne, Switzerland; [2]Departments of Neurobiology and Statistics, University of Chicago, Chicago, United States; [3]Grossman Center for Quantitative Biology and Human Behavior, University of Chicago, Chicago, United States

**Abstract** In adult dentate gyrus neurogenesis, the link between maturation of newborn neurons and their function, such as behavioral pattern separation, has remained puzzling. By analyzing a theoretical model, we show that the switch from excitation to inhibition of the GABAergic input onto maturing newborn cells is crucial for their proper functional integration. When the GABAergic input is excitatory, cooperativity drives the growth of synapses such that newborn cells become sensitive to stimuli similar to those that activate mature cells. When GABAergic input switches to inhibitory, competition pushes the configuration of synapses onto newborn cells toward stimuli that are different from previously stored ones. This enables the maturing newborn cells to code for concepts that are novel, yet similar to familiar ones. Our theory of newborn cell maturation explains both how adult-born dentate granule cells integrate into the preexisting network and why they promote separation of similar but not distinct patterns.

*For correspondence:
gozel@uchicago.edu

Present address: †Departments of Neurobiology and Statistics, University of Chicago, Chicago, United States

Competing interests: The authors declare that no competing interests exist.

## Introduction

In the adult mammalian brain, neurogenesis, the production of new neurons, is restricted to a few brain areas, such as the olfactory bulb and the dentate gyrus (*Deng et al., 2010*). The dentate gyrus is a major entry point of input from cortex, primarily entorhinal cortex (EC), to the hippocampus (*Amaral et al., 2007*), which is believed to be a substrate of learning and memory (*Jarrard, 1993*). Adult-born cells in dentate gyrus mostly develop into dentate granule cells (DGCs), the main excitatory cells that project to area CA3 of hippocampus (*Deng et al., 2010*).

The properties of rodent adult-born DGCs change as a function of their maturation stage, until they become indistinguishable from other mature DGCs at approximately 8 weeks (*Deng et al., 2010*; *Johnston et al., 2016*; *Figure 1a*). Many of them die before they fully mature (*Dayer et al., 2003*). Their survival is experience dependent and relies on NMDA receptor activation (*Tashiro et al., 2006*). Initially, newborn DGCs have enhanced excitability (*Schmidt-Hieber et al., 2004*; *Li et al., 2017*) and stronger synaptic plasticity than mature DGCs, reflected by a larger long-term potentiation (LTP) amplitude and a lower threshold for induction of LTP (*Wang et al., 2000*; *Schmidt-Hieber et al., 2004*; *Ge et al., 2007*). Furthermore, after 4 weeks of maturation adult-born DGCs have only weak connections to interneurons, while at 7 weeks of age, their activity causes indirect inhibition of mature DGCs (*Temprana et al., 2015*).

Newborn DGCs receive no direct connections from mature DGCs (*Deshpande et al., 2013*; *Alvarez et al., 2016*) (yet see *Vivar et al., 2012*), but are indirectly activated via interneurons (*Alvarez et al., 2016*; *Heigele et al., 2016*). At about 3 weeks after birth, the γ-aminobutyric acid (GABAergic) input from interneurons to adult-born DGCs switches from excitatory in the early phase to inhibitory in the late phase of maturation (*Ge et al., 2006*; *Deng et al., 2010*) ('GABA-switch', *Figure 1a*). Analogous to a similar transition during embryonic and early postnatal stages

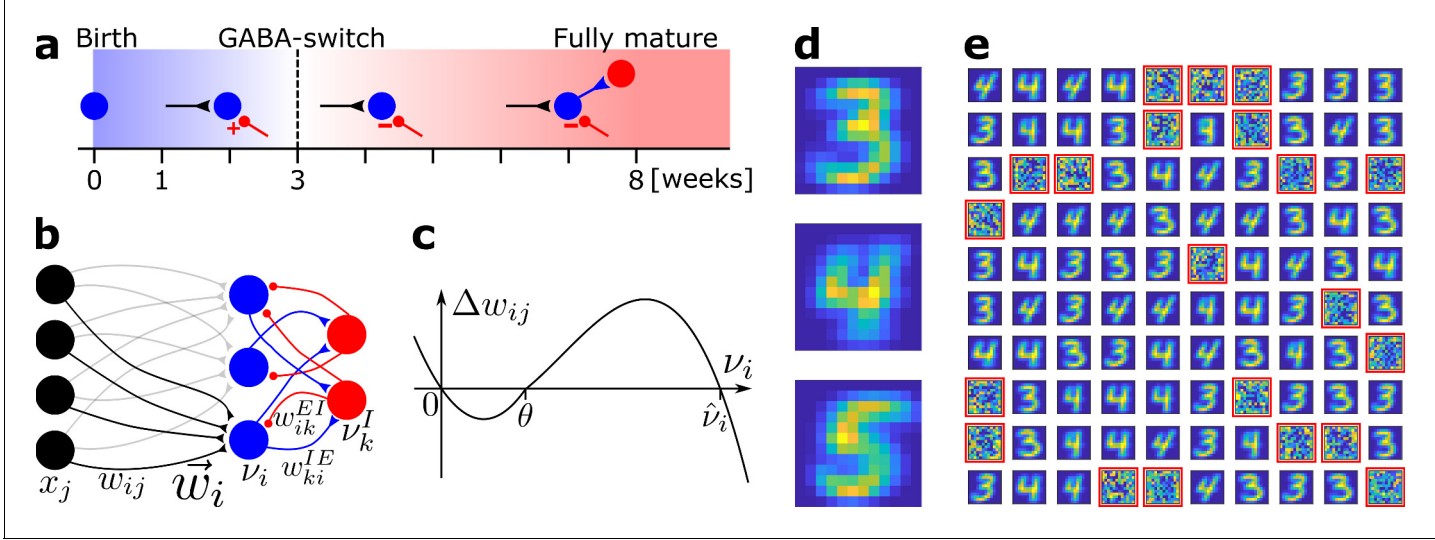

**Figure 1.** Network model and pretraining. (**a**) Integration of an adult-born DGC (blue) as a function of time: GABAergic synaptic input (red) switches from excitatory (+) to inhibitory (−); strong connections to interneurons develop only later; glutamatergic synaptic input (black), interneuron (red). (**b**) Network structure. EC neurons (black, rate $x_j$) are fully connected with weights $w_{ij}$ to DGCs (blue, rate $\nu_i$). The feedforward weight vector $\vec{w}_i$ onto neuron $i$ is depicted in black. DGCs and interneurons (red, rate $\nu_k^I$) are mutually connected with probability $p_{IE}$ and $p_{EI}$ and weights $w_{ki}^{IE}$ and $w_{ik}^{EI}$, respectively. Connections with a triangular (round) end are glutamatergic (GABAergic). (**c**) Given presynaptic activity $x_j > 0$, the weight update $\Delta w_{ij}$ is shown as a function of the firing rate $\nu_i$ of the postsynaptic DGC with LTD for $\nu_i < \theta$ and LTP for $\theta < \nu_i < \hat{\nu}_i$. (**d**) Center of mass for three ensembles of patterns from the MNIST data set, visualized as 12 × 12 pixel patterns. The two-dimensional arrangements and colors are for visualization only. (**e**) One hundred receptive fields, each defined as the set of feedforward weights, are represented in a two-dimensional organization. After pretraining with patterns from MNIST digits 3 and 4, 79 DGCs have receptive fields corresponding to threes and fours of different writing styles, while 21 remain unselective (highlighted by red frames).

The online version of this article includes the following figure supplement(s) for figure 1:

**Figure supplement 1.** Dentate gyrus network.

(*Wang and Kriegstein, 2011*), the GABA-switch is caused by a change in the expression profile of chloride cotransporters. In the early phase of maturation, newborn cells express the $Na^+ - K^+ - 2Cl^-$ cotransporter NKCC1, which leads to a high intracellular chloride concentration. Hence, the GABA reversal potential is higher than the resting potential (*Ge et al., 2006*; *Heigele et al., 2016*), and GABAergic inputs lead to $Cl^-$ ions outflow through the $GABA_A$ ionic receptors, which results in depolarization of the newborn cell (*Ben-Ari, 2002*; *Owens and Kriegstein, 2002*). In the late phase of maturation, expression of the $K^+ - Cl^-$-coupled cotransporter KCC2 kicks in, which lowers the intracellular chloride concentration of the newborn cell to levels similar to those of mature cells, leading to a hyperpolarization of the cell membrane due to $Cl^-$ inflow upon GABAergic stimulation (*Ben-Ari, 2002*; *Owens and Kriegstein, 2002*). The transition from depolarizing (excitatory) to hyperpolarizing (inhibitory) effects of GABA is referred to as the 'GABA-switch'. It has been shown that GABAergic inputs are crucial for the integration of newborn DGCs into the preexisting circuit (*Ge et al., 2006*; *Chancey et al., 2013*; *Alvarez et al., 2016*; *Heigele et al., 2016*).

The mammalian dentate gyrus contains – just like hippocampus in general – a myriad of inhibitory cell types (*Freund and Buzsáki, 1996*; *Somogyi and Klausberger, 2005*; *Klausberger and Somogyi, 2008*), including basket cells, chandelier cells, and hilar cells (*Figure 1—figure supplement 1*). Basket cells can be subdivided in two categories: some express cholecystokinin (CCK) and vasoactive intestinal polypeptide (VIP), while the others express parvalbumin (PV) and are fast-spiking (*Freund and Buzsáki, 1996*; *Amaral et al., 2007*). Chandelier cells also express PV (*Freund and Buzsáki, 1996*). Overall, it has been estimated that PV is expressed in 15–21% of all dentate GABAergic cells (*Freund and Buzsáki, 1996*) and in 20–25% of the GABAergic neurons in the granule cell layer (*Houser, 2007*). Amongst the GABAergic hilar cells, 55% express somatostatin (SST) (*Houser, 2007*) and somatostatin-positive interneurons (SST-INs) represent about 16% of the GABAergic neurons in the dentate gyrus as a whole (*Freund and Buzsáki, 1996*). While axons of

hilar interneurons stay in the hilus and provide perisomatic inhibition onto dentate GABAergic cells, axons of hilar-perforant-path-associated interneurons (HIPP) extend to the molecular layer and provide dendritic inhibition onto both DGCs and interneurons (*Yuan et al., 2017*). HIPP axons generate lots of synaptic terminals and extend as far as 3.5 mm along the septotemporal axis of the dentate gyrus (*Amaral et al., 2007*). PV-expressing interneurons (PV-INs) and SST-INs both target adult-born DGCs early (after 2–3 weeks) in their maturation (*Groisman et al., 2020*). PV-INs provide both feedforward inhibition and feedback inhibition (also called lateral inhibition) to the DGCs (*Groisman et al., 2020*). In general, SST-INs provide lateral, but not feedforward, inhibition onto DGCs (*Stefanelli et al., 2016*; *Groisman et al., 2020*; *Figure 1—figure supplement 1*).

Adult-born DGCs are preferentially reactivated by stimuli similar to the ones they experienced during their early phase of maturation, up to 3 weeks after cell birth (*Tashiro et al., 2007*). Even though the amount of newly generated cells per month is rather low (3–6% of the total DGCs population [*van Praag et al., 1999*; *Cameron and McKay, 2001*]), adult-born DGCs are critical for behavioral pattern separation (*Clelland et al., 2009*; *Sahay et al., 2011a*; *Jessberger et al., 2009*), in particular in tasks where similar stimuli or contexts have to be discriminated (*Clelland et al., 2009*; *Sahay et al., 2011a*). However, the functional role of adult-born DGCs is controversial (*Sahay et al., 2011b*; *Aimone et al., 2011*). One view is that newborn DGCs contribute to pattern separation through a modulatory role (*Sahay et al., 2011b*). Another view suggests that newborn DGCs act as encoding units that become sensitive to features of the environment which they encounter during a critical window of maturation (*Kee et al., 2007*; *Tashiro et al., 2007*). Some authors have even challenged the role of newborn DGCs in pattern separation in the classical sense and have proposed a pattern integration effect instead (*Aimone et al., 2011*), while others suggest a dynamical (*Aljadeff et al., 2015*; *Shani-Narkiss et al., 2020*) or forgetting (*Akers et al., 2014*) role for newborn DGCs. Within that broader controversy, we ask two specific questions: First, why are GABAergic inputs crucial for the integration of newborn DGCs into the preexisting circuit? And second, why are newborn DGCs particularly important in tasks where similar stimuli or contexts have to be discriminated?

To address these questions, we present a model of how newborn DGCs integrate into the preexisting circuit. In contrast to earlier models where synaptic input connections onto newborn cells were assumed to be strong enough to drive them (*Chambers et al., 2004*; *Becker, 2005*; *Crick and Miranker, 2006*; *Wiskott et al., 2006*; *Chambers and Conroy, 2007*; *Aimone et al., 2009*; *Appleby and Wiskott, 2009*; *Weisz and Argibay, 2009*; *Temprana et al., 2015*; *Finnegan and Becker, 2015*; *DeCostanzo et al., 2018*), our model uses an unsupervised biologically plausible Hebbian learning rule that makes synaptic connections between EC and newborn DGCs either disappear or grow from small values at birth to values that eventually enable feedforward input from EC to drive DGCs. Contrary to previous modeling studies, our plasticity model does not require an artificial renormalization of synaptic connection weights since model weights are naturally bounded by the synaptic plasticity rule. We show that learning with a biologically plausible plasticity rule is possible thanks to the GABA-switch, which has been overlooked in previous modeling studies. Specifically, the growth of synaptic weights from small values is supported in our model by the excitatory action of GABA, whereas, after the switch, specialization of newborn cells arises from competition between DGCs, triggered by the inhibitory action of GABA. Furthermore, our theory of adult-born DGCs integration yields a transparent explanation of why newborn cells favor pattern separation of similar stimuli, but do not impact pattern separation of distinct stimuli.

## Results

We model a small patch of cells within dentate gyrus as a recurrent network of 100 DGCs and 25 GABAergic interneurons, omitting the Mossy cells for the sake of simplicity (*Figure 1b*). The modeled interneurons correspond to SST-INs from the HIPP category, as they are the providers of feedback inhibition to DGCs through dendritic projections (*Stefanelli et al., 2016*; *Yuan et al., 2017*; *Groisman et al., 2020*; *Figure 1—figure supplement 1*). The activity of a DGC with index $i$ and an interneuron with index $k$ is described by their continuous firing rates $\nu_i$ and $\nu_k^I$, respectively. Firing rates are modeled by neuronal frequency–current curves that vanish for weak input and increase if the total input into a neuron is larger than a firing threshold. Since newborn DGCs exhibit enhanced excitability early in maturation (*Schmidt-Hieber et al., 2004*; *Li et al., 2017*), the firing threshold of

model neurons increases during maturation from a lower to a higher value (Materials and methods). Connectivity in a localized patch of dentate neurons is high: DGCs densely project to GABAergic interneurons (*Acsády et al., 1998*), and SST-INs heavily project to cells in their neighborhood (*Amaral et al., 2007*). Hence, in the recurrent network model, each model DGC projects to, and receives input from, a given interneuron with probability 0.9. The exact percentage of GABAergic neurons (or SST-INs) in the dentate gyrus as a whole is not known, but has been estimated at about 10% and only a fraction of these are SST-INs (*Freund and Buzsáki, 1996*). The number of inhibitory neurons in our model network might therefore seem too high. However, our results are robust to substantial changes in the number of inhibitory neurons (*Supplementary file 2*).

Each of the 100 model DGCs receives input from a set of 144 model EC cells (*Figure 1b*). In the rat, the number of DGCs has been estimated to be about $10^6$, while the number of EC input cells is estimated to be about $2 \cdot 10^5$ (*Andersen et al., 2007*), yielding an expansion factor from EC to dentate gyrus of about 5. Theoretical analysis suggests that the expansion of the number of neurons enhances decorrelation of the representation of input patterns (*Marr, 1969*; *Albus, 1971*; *Marr, 1971*; *Rolls and Treves, 1998*) and promotes pattern separation (*Babadi and Sompolinsky, 2014*). Our standard network model does not reflect this expansion because we want to highlight the particular ability of adult neurogenesis in combination with the GABA-switch to decorrelate input patterns independently of specific choices of the network architecture. However, we show later that an enlarged network with an expansion from 144 model EC cells to 700 model DGCs (similar to the anatomical expansion factor) yields similar results.

At birth, a DGC with index $i$ does not receive synaptic glutamatergic input yet. Hence, the connection from any model EC cell with index $j$ is initialized at $w_{ij} = 0$. The growth or decay of the synaptic strength $w_{ij}$ of the connection from $j$ to $i$ is controlled by a Hebbian plasticity rule (*Figure 1c*):

$$\Delta w_{ij} = \eta \left\{ \gamma x_j \nu_i [\nu_i - \theta]_+ - \alpha x_j \nu_i [\theta - \nu_i]_+ - \beta w_{ij} [\nu_i - \theta]_+ \nu_i^3 \right\} \qquad (1)$$

where $x_j$ is the firing rate of the presynaptic EC neuron, $\eta$ ('learning rate') is the susceptibility of a cell to synaptic plasticity, and $\alpha, \beta, \gamma$ are positive parameters (Materials and methods, *Table 1*). The first term on the right-hand side of *Equation (1)* describes LTP whenever the presynaptic neuron is active ($x_j > 0$) and the postsynaptic firing $\nu_i$ is above a threshold $\theta$; the second term on the right-hand side of *Equation (1)* describes long-term depression (LTD) whenever the presynaptic neuron is active and the postsynaptic firing rate is positive but below the threshold $\theta$; LTD stops if the synaptic weight is zero. Such a combination of LTP and LTD is consistent with experimental data (*Artola et al., 1990*; *Sjöström et al., 2001*) as shown in earlier rate-based (*Bienenstock et al., 1982*) or spike-based (*Pfister and Gerstner, 2006*) plasticity models. The third term on the right-hand side of *Equation (1)* implements heterosynaptic plasticity (*Chistiakova et al., 2014*;

**Table 1.** Parameters for the simulations.

|  | Biologically plausible network | | Simplified network | |
|---|---|---|---|---|
| Network | $N_{EC} = 144$ | $N_I = 25$ | $N_{EC} = 128$ | $N_{DGC} = 3$ |
|  | $N_{DGC} = 100$ (*Figures 1–4*) | | | |
|  | $N_{DGC} = 700$ (*Figure 4—figure supplement 1–2*) | | | |
| Connectivity | $w_{IE} = 1$ | $w_{EI} = -\frac{1}{p_{EI} N_I}$ | $w_{rec} = -1.2$ | |
|  | $p_{IE} = 0.9$ | $p_{EI} = 0.9$ | | |
| Dynamics | $\tau_m = 20$ ms | $\tau_{inh} = 2$ ms | $\tau_m = 20$ ms | |
|  | $L = 0.5$ | $p^* = 0.1$ | | |
| Plasticity | $\alpha_0 = 0.05$ | $\beta = 1$ | $\alpha_0 = 0.03$ | $\beta = 1$ |
|  | $\gamma_0 = 10$ | $\theta = 0.15$ | $\gamma_0 = 1.65$ | $\theta = 0.15$ |
|  | $\nu_0 = 0.2$ | $\gamma = 9.85$ | $\gamma = 1.5$ | |
| Numerical simulations | $\Delta t = 0.1$ ms | $\eta = 0.01$ | $\Delta t = 1$ ms | $\eta = 0.01$ |
|  | $\eta_b = 0.01$ | | | |

*Zenke and Gerstner, 2017*): whenever strong presynaptic input arriving at synapses $k \neq j$ drives the firing of postsynaptic neuron $i$ at a rate above θ, the weight of a synapse $j$ is downregulated if synapse $j$ does not receive any input, while the weights of synapses $k \neq j$ are simultaneously increased due to the first term (*Lynch et al., 1977*). Importantly, the threshold condition for the third term (postsynaptic rate above θ) is the same as that for induction of LTP in the first term so that if some synapses are potentiated, silent synapses are depressed. In the model, heterosynaptic interaction between synapses is induced since information about postsynaptic activity is shared across synapses. This could be achieved in biological neurons via backpropagating action potentials or similar depolarization of the postsynaptic membrane potential at several synaptic locations; alternatively, heterosynaptic crosstalk could be implemented by signaling molecules. Note that since our neuron model is a point neuron, all synapses are neighbors of each other. In our model, the 'heterosynaptic' term has a negative sign which ensures that the weights cannot grow without bounds (Materials and methods). In this sense, the third term has a 'homeostatic' function (*Zenke and Gerstner, 2017*), yet acts on a time scale faster than experimentally observed homeostatic synaptic plasticity (*Turrigiano et al., 1998*).

We ask whether such a biologically plausible plasticity rule enables adult-born DGCs to be integrated in an existing network of mature cells. To address this question, we exploit two observations (*Figure 1a*): first, the effect of interneurons onto newborn DGCs exhibits a GABA-switch from excitatory to inhibitory after about three weeks of maturation (*Ge et al., 2006*; *Deng et al., 2010*) and, second, newborn DGCs receive input from interneurons early in their maturation (before the third week), but project back to interneurons only later (*Temprana et al., 2015*). For simplicity, no plasticity rule was implemented within the dentate gyrus: connections between newborn DGCs and inhibitory cells are either absent or present with a fixed value (see below). However, before integration of adult-born DGCs can be addressed, an adult-stage network where mature cells already store some memories has to be constructed.

## Mature neurons represent prototypical input patterns

In an adult-stage network, some mature cells already have a functional role. Hence, we start with a network that already has strong random EC-to-DGC connection weights (Materials and methods). We then pretrain our network of 100 DGCs using the same learning rule (*Equation (1)*, with identical learning rate η for all DGCs) that we will use later for the integration of newborn cells. For the stimulation of EC cells, we apply patterns representing thousands of handwritten digits in different writing styles from MNIST, a standard data set in artificial intelligence (*Lecun et al., 1998*). Even though we do not expect EC neurons to show a two-dimensional arrangement, the use of two-dimensional patterns provides a simple way to visualize the activity of all 144 EC neurons in our model (*Figure 1d*). We implicitly model feedforward inhibition from PV-INs (*Groisman et al., 2020*; *Figure 1—figure supplement 1*) by normalizing input patterns so that all inputs have the same amplitude (Materials and methods). Below, we present results for a representative combination of three digits (digits 3, 4, and 5), but other combinations of digits have also been tested (*Supplementary file 1*).

After pretraining with patterns from digits 3 and 4 in a variety of writing styles, we examine the receptive field of each DGC. Each receptive field, consisting of the connections from all 144 EC neurons onto one DGC, is characterized by its spatial structure (i.e. the pattern of connection weights) and its total strength (i.e. the efficiency of the optimal stimulus to drive the cell). We observe that out of the 100 DGCs, some have developed spatial receptive fields that correspond to different writing styles of digit 3, others receptive fields that correspond to variants of digit 4 (*Figure 1e*).

Behavioral discrimination has been shown to be correlated with classification accuracy based on DGC population activity (*Woods et al., 2020*). Hence, to quantify the representation quality, we compute classification performance by a linear classifier that is driven by the activity of our 100 DGC model cells (Materials and methods). At the end of pretraining, the classification performance for patterns of digits 3 and 4 from a distinct test set not used during pretraining is high: 99.25% (classification performance on digit 3: 98.71%; digit 4: 99.80%), indicating that nearly all input patterns of the two digits are well represented by the network of mature DGCs. The median classification performance for 10 random combinations of two groups of pretrained digits is 98.54%, the 25th percentile 97.26%, and the 75th percentile 99.5% (*Supplementary file 1*).

A detailed mathematical analysis (Materials and methods) shows that heterosynaptic plasticity in *Equation (1)* ensures that the total strength of the receptive field of each selective DGC converges

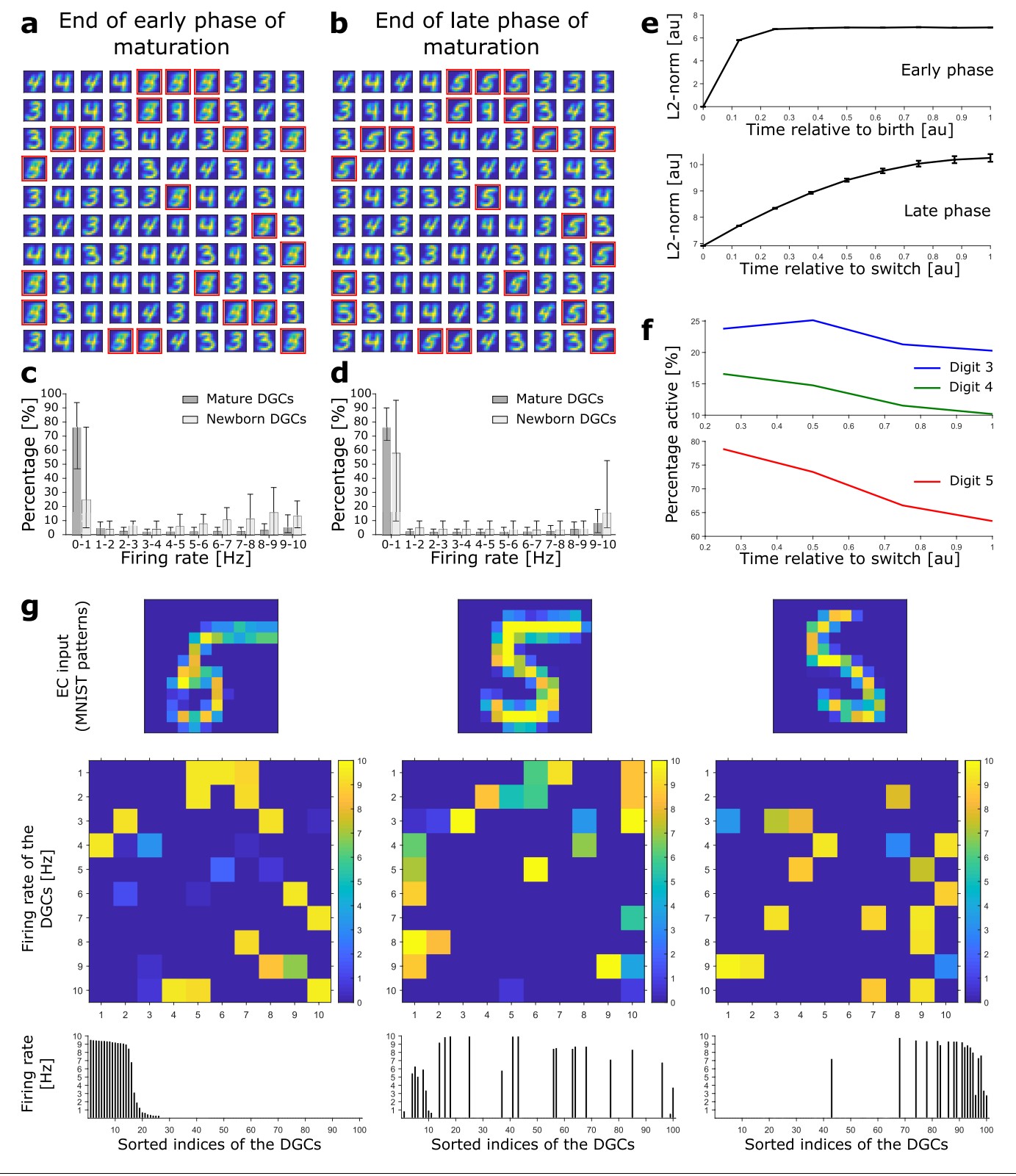

**Figure 2.** Newborn DGCs become selective for novel patterns during maturation. (**a**) Unselective neurons are replaced by newborn DGCs, which learn their feedforward weights while patterns from digits 3, 4, and 5 are presented. At the end of the early phase of maturation, the receptive fields of all newborn DGCs (red frames) show mixed selectivity. (**b**) At the end of the late phase of maturation, newborn DGCs are selective for patterns from the novel digit 5, with different writing styles. (**c, d**) Distribution of the percentage of model DGCs (mean with 10th and 90th percentiles) in each firing rate

*Figure 2 continued on next page*

*Figure 2 continued*

bin at the end of the early (**c**) and late (**d**) phase of maturation. Statistics calculated across MNIST patterns ('3's, '4's, '5's). Percentages are per subpopulation (mature and newborn). Note that neurons with firing rate < 1 Hz for one pattern may fire at medium or high rate for another pattern. (**e**) The L2-norm of the feedforward weight vector onto newborn DGCs (mean ± SEM) increases as a function of maturation indicating growth of synapses and receptive field strength. Horizontal axis: time = 1 indicates end of early (top) or late (bottom) phase (two epochs per phase, $\eta = 0.0005$). (**f**) Percentage of newborn DGCs activated (firing rate > 1 Hz) by a stimulus averaged over test patterns of digits 3, 4, and 5 as a function of maturation. (**g**) At the end of the late phase of maturation, three different patterns of digit 5 applied to EC neurons (top) cause different firing rate patterns of the 100 DGCs arranged in a matrix of 10-by-10 cells (middle). DGCs with a receptive field (see **b**) similar to a presented EC activation pattern respond more strongly than the others. Bottom: Firing rates of the DGCs with indices sorted from highest to lowest firing rate in response to the first pattern. All three patterns shown come from the testing set, and are correctly classified using our readout network.

The online version of this article includes the following figure supplement(s) for figure 2:

**Figure supplement 1.** Activity of 100 model DGCs in response to different patterns.

**Figure supplement 2.** Receptive fields of DGCs.

to a stable value which is similar for selective DGCs confirming the homeostatic function of heterosynaptic plasticity (*Zenke and Gerstner, 2017*). As a consequence, synaptic weights are intrinsically bounded without the need to impose hard bounds on the weight dynamics. Moreover, we find that the spatial structure of the receptive field represents the weighted average of all those input

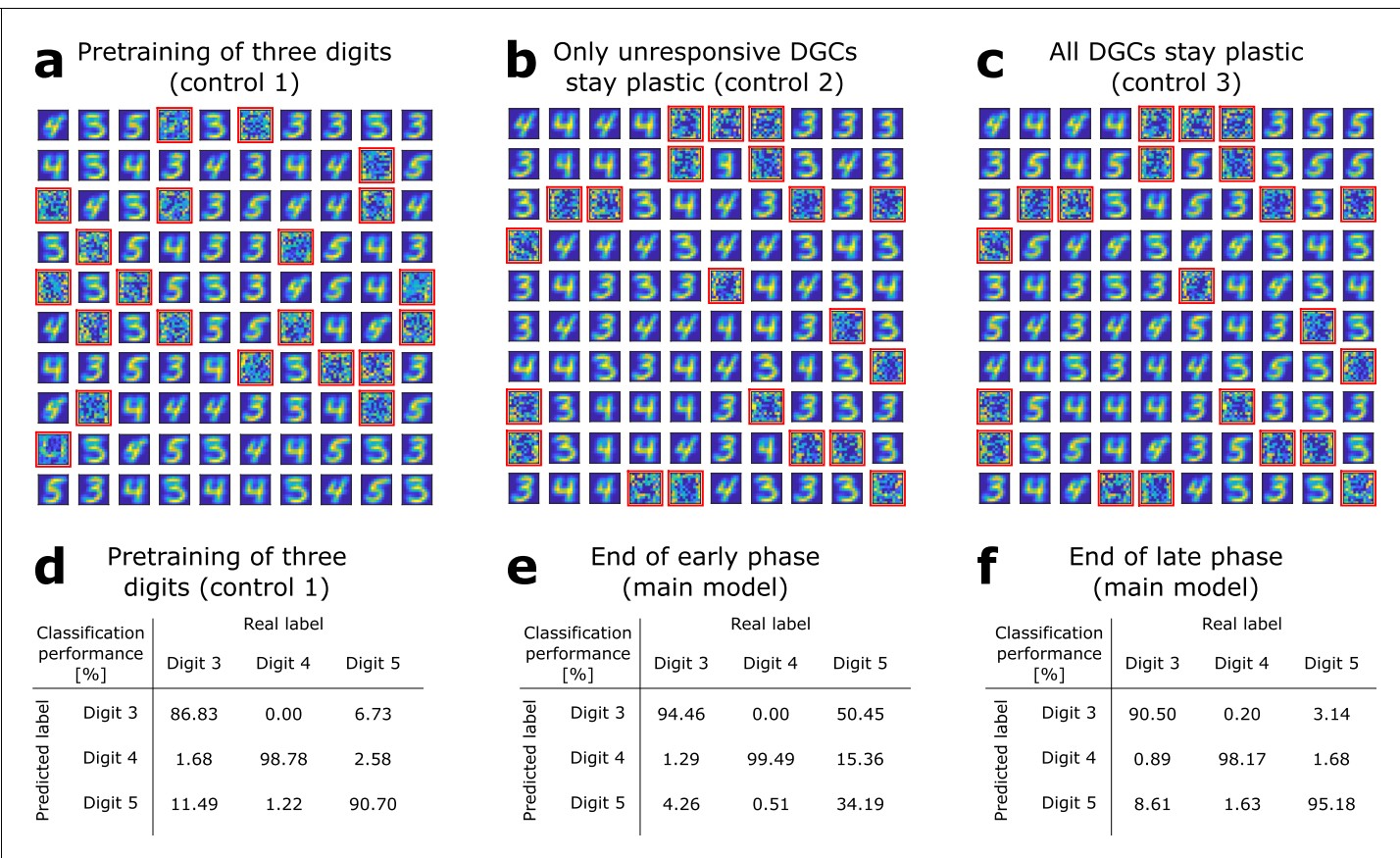

**Figure 3.** The GABA-switch guides learning of novel representations. (**a**) Pretraining on digits 3, 4, and 5 simultaneously without neurogenesis (control 1). Patterns from digits 3, 4, and 5 are presented to the network while all DGCs learn their feedforward weights. After pretraining, 79 DGCs have receptive fields corresponding to the three learned digits, while 21 remain unselective (as in *Figure 1e*). (**b**) Sequential training without neurogenesis (control 2). After pretraining as in *Figure 1e*, the unresponsive neurons stay plastic, but they fail to become selective for digit 5 when patterns from digits 3, 4, and 5 are presented in random order. (**c**) Sequential training without neurogenesis but all DGCs stay plastic (control 3). Some of the DGCs previously responding to patterns from digits 3 or 4 become selective for digit 5. (**d–f**) Confusion matrices. Classification performance in percent (using a linear classifier as readout network) for control 1 (**d**) and for the main model at the end of the early (**e**) and late (**f**) phase; *Figure 2a,b*.

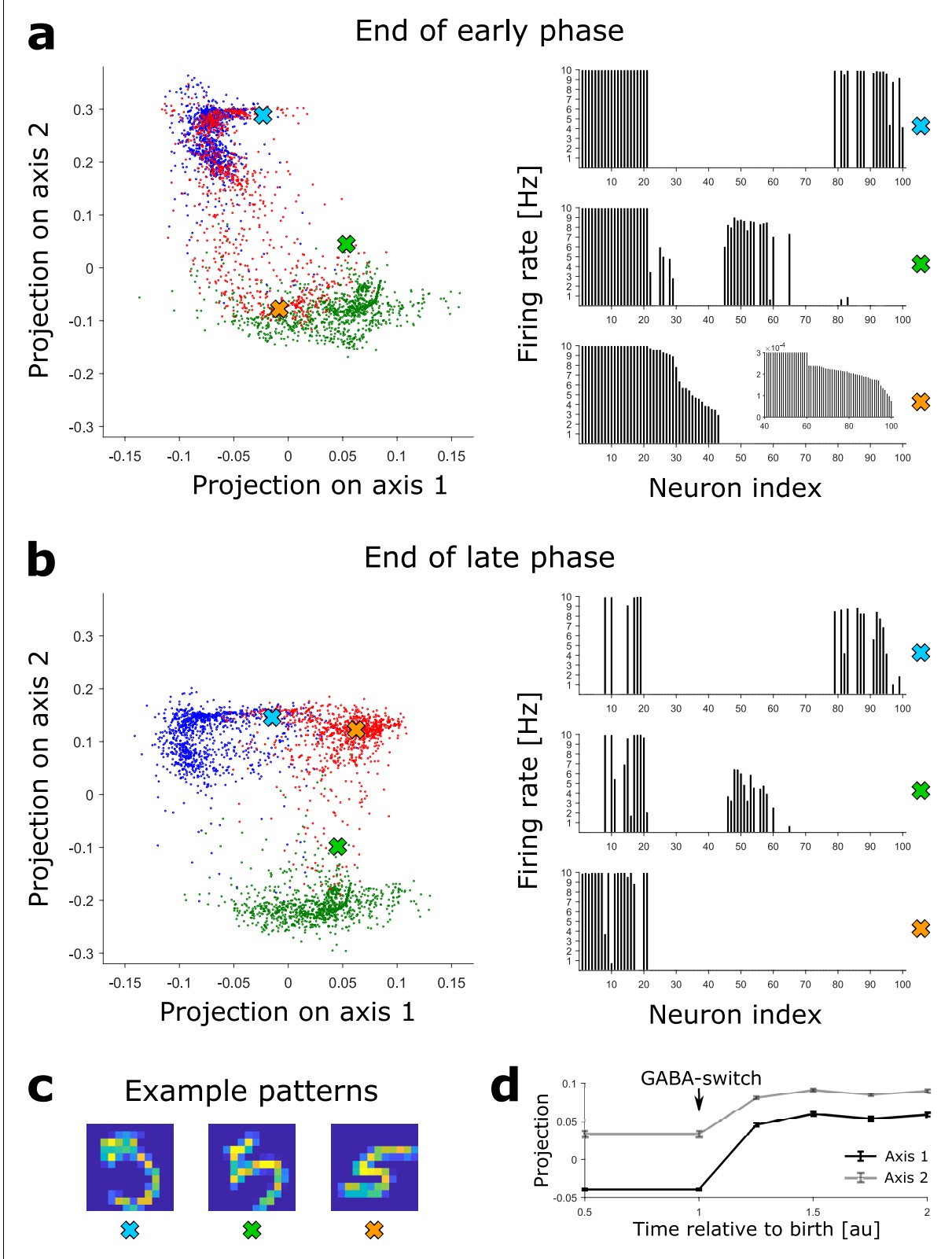

**Figure 4.** Novel patterns expand the representation into a previously empty subspace. (**a**) Left: The DGC activity responses at the end of the early phase of maturation of newborn DGCs are projected on discriminatory axes. Each point corresponds to the representation of one input pattern. Color indicates digit 3 (blue), 4 (green), and 5 (red). Right: Firing rate profiles of three example patterns (highlighted by crosses on the left) are sorted from high to low for the pattern represented by the orange cross (inset: zoom of firing rates of DGCs with low activity). (**b**) Same as (a), but at the end of the

*Figure 4 continued on next page*

*Figure 4 continued*

late phase of maturation of newborn DGCs. Note that the red dots around the orange cross have moved into a different subspace. (c) Example patterns of digit 5 corresponding to the symbols in (a) and (b). All three are accurately classified by our readout network. (d) Evolution of the mean (± SEM) of the projection of the activity upon presentation of all test patterns of digit 5.

The online version of this article includes the following figure supplement(s) for figure 4:

**Figure supplement 1.** Receptive fields of the DGCs in a larger network with $N_{DGC} = 700$ (all other parameters unchanged).

**Figure supplement 2.** Receptive fields of the DGCs in a larger network with $N_{DGC} = 700$ (all other parameters unchanged), when only a fraction of unresponsive units are replaced by newborn DGCs.

patterns for which that DGC is responsive. The mathematical analysis also shows that those DGCs that do not develop selectivity have weak synaptic connections and a very low total strength of the receptive field.

After convergence of synaptic weights during pretraining, selective DGCs are considered mature cells. Mature cells are less plastic than newborn cells (*Schmidt-Hieber et al., 2004*; *Ge et al., 2007*). So in the following, unless specified otherwise, we set $\eta = 0$ in *Equation (1)* for mature cells (feedforward connection weights from EC to mature cells remain therefore fixed). A scenario where mature cells retain synaptic plasticity is also investigated (see Robustness of the model and *Supplementary file 4*). Some DGCs did not develop any strong weight patterns during pretraining and exhibit unselective receptive fields (highlighted in red in *Figure 1e*). We classify these as unresponsive units.

## Newborn neurons become selective for novel patterns during maturation

In our main neurogenesis model, we replace unresponsive model units by plastic newborn DGCs ($\eta > 0$ in *Equation (1)*), which receive lateral GABAergic input but do not receive feedforward input yet (all weights from EC are set to zero). The replacement of unresponsive neurons reflects the fact that unresponsive units have weak synaptic connections and, experimentally, a lack of NMDA receptor activation has been shown to be deleterious for the survival of newborn DGCs (*Tashiro et al., 2006*). To mimic exposure of an animal to a novel set of stimuli, we now add input patterns from digit 5 to the set of presented stimuli, which was previously limited to patterns of digits 3 and 4. The novel patterns from digit 5 are randomly interspersed into the sequence of patterns from digits 3 and 4; in other words, the presentation sequence was not optimized with a specific goal in mind.

We postulate that functional integration of newborn DGCs requires the two-step maturation process caused by the GABA-switch from excitation to inhibition. Since excitatory GABAergic input potentially increases correlated activity within the dentate gyrus network, we predict that newborn DGCs respond to familiar stimuli during the early phase of maturation, but not during the late phase, when inhibitory GABAergic input leads to competition.

To test this hypothesis, our model newborn DGCs go through two maturation phases (Materials and methods). The early phase of maturation is cooperative because, for each pattern presentation, activated mature DGCs indirectly excite the newborn DGCs via GABAergic interneurons. We assume that in natural settings, the activation of $GABA_A$ receptors is low enough that the mean membrane potential remains below the chloride reversal potential at which shunting inhibition would be induced (*Heigele et al., 2016*). In this regime, the net effect of synaptic activity is hence excitatory. This lateral activation of newborn DGCs drives the growth of their receptive fields in a direction similar to those of the currently active mature DGCs. Consistent with our hypothesis we find that, at the end of the early phase of maturation, newborn DGCs show a receptive field corresponding to a mixture of several input patterns (*Figure 2a*).

In the late phase of maturation, model newborn DGCs receive inhibitory GABAergic input from interneurons, similar to the input received by mature DGCs. Given that at the end of the early phase, newborn DGCs have receptive fields similar to those of mature DGCs, lateral inhibition induces competition with mature DGCs for activation during presentation of patterns from the novel digit. Because model newborn DGCs start their late phase of maturation with a higher excitability (lower threshold) compared to mature DGCs, consistent with observed enhanced excitability of newborn cells (*Schmidt-Hieber et al., 2004*; *Li et al., 2017*), the activation of newborn DGCs is facilitated for those input patterns for which no mature DGC has preexisting selectivity. Therefore, in the late

phase of maturation, competition drives the synaptic weights of most newborn DGCs toward receptive fields corresponding to different subcategories of the ensemble of input patterns of the novel digit 5 (*Figure 2b*).

The total strength of the receptive field of a given DGC can be characterized by the sum of the squared synaptic weights of all feedfoward projections onto the cell (i.e. the square of the L2-norm). During maturation, the L2-norm of the feedforward weights onto newborn DGCs increases (*Figure 2e*) indicating an increase in total glutamatergic innervation, e.g., through an increase in the number and size of spines (*Zhao et al., 2006*). Nevertheless, the distribution of firing rates of newborn DGCs is shifted to lower values at the end of the late phase compared to the end of the early phase of maturation (*Figure 2c,d*), consistent with in vivo calcium imaging recordings showing that newborn DGCs are more active than mature DGCs (*Danielson et al., 2016*).

We emphasize that upon presentation of a pattern of a given digit, only those DGCs with a receptive field similar to the specific writing style of the presented pattern become strongly active, others fire at a medium firing rate, yet others at a low rate (*Figure 2g*). As a consequence, the firing rate of a particular newborn DGC at the end of its maturation to a pattern from digit 5 is strongly modulated by the specific choice of stimulation pattern within the class of '5's. Analogous results are obtained for patterns from pretrained digits 3 and 4 (*Figure 2—figure supplement 1*). Hence, the ensemble of DGCs is effectively performing pattern separation *within* each digit class as opposed to a simple ternary classification task. The selectivity of newborn DGCs develops during maturation. Indeed, during the late, competitive, phase, the percentage of active newborn DGCs decreases, both upon presentation of familiar patterns (digits 3 and 4), as well as upon presentation of novel patterns (digit 5) (*Figure 2f*). This reflects the development of the selectivity of our model newborn DGCs from broad to narrow tuning, consistent with experimental observations (*Marín-Burgin et al., 2012*; *Danielson et al., 2016*).

If two novel ensembles of digits (instead of a single one) are introduced during maturation of newborn DGCs, we observe that some newborn DGCs become selective for one of the novel digits, while others become selective for the other novel digit (*Figure 2—figure supplement 2*). This was expected, since we have found earlier that DGCs are becoming selective for different prototype writing styles even *within* a digit category; hence introducing several additional digit categories of novel patterns simply increases the prototype diversity. Therefore, newborn DGCs can ultimately promote separation of several novel overarching categories of patterns, no matter if they are learned simultaneously or sequentially (*Figure 2—figure supplement 2*).

## Adult-born neurons promote better discrimination

As above, we compute classification performance of our model network as a surrogate for behavioral discrimination (*Woods et al., 2020*). At the end of the late phase of maturation of newborn DGCs, we obtain an overall classification performance of 94.56% for the three ensembles of digits (classification performance for digit 3: 90.50%; digit 4: 98.17%; digit 5: 95.18%). Confusion matrices show that although novel patterns are not well classified at the end of the early phase of maturation (*Figure 3e*), they are as well classified as pretrained patterns at the end of the late phase of maturation (*Figure 3f*).

We compare this performance with that of a network where all three digit ensembles are directly simultaneously pretrained starting from random weights (*Figure 3a*, control 1). In this case, the overall classification performance is 92.09% (classification performance for digit 3: 86.83%; digit 4: 98.78%; digit 5: 90.70%). The confusion matrix shows that all three digits are decently classified, but with an overall lower performance (*Figure 3d*). Across 10 simulation experiments, classification performance is significantly higher when a novel ensemble of patterns is learned sequentially by newborn DGCs ($P_2$; *Supplementary file 1*), than if all patterns are learned simultaneously ($P_1$; *Supplementary file 1*). Indeed, the distribution of $P_2 - P_1$ for the 10 simulation experiments has a mean which is significantly different from zero (Wilcoxon signed rank test: p-val = 0.0020, Wilcoxon signed rank = 55; one-way t-test: p-val = 0.0269, t-stat = 2.6401, df = 9; *Supplementary file 1*).

## The GABA-switch guides learning of novel representations

To assess whether maturation of newborn DGCs promotes learning of a novel ensemble of digit patterns, we compare our results with two control models without neurogenesis (controls 2 and 3).

In control 2, similar to the neurogenesis case, the feedforward weights and thresholds of mature DGCs are fixed (learning rate $\eta = 0$) after pretraining with patterns from digits 3 and 4, while the thresholds and weights of all unresponsive neurons remain plastic ($\eta > 0$) upon introduction of patterns from the novel digit 5. The only differences to the model with neurogenesis are that unresponsive neurons: (1) keep their feedforward weights (i.e. no reinitialization to zero values) and (2) keep the same connections from and to inhibitory neurons. In this case, we find that the previously unresponsive DGCs do not become selective for the novel digit 5, no matter during how many epochs patterns are presented (we went up to 100 epochs) (*Figure 3b*, control 2). Therefore, if patterns from digit 5 are presented to the network, the model fails to discriminate them from the previously learned digits 3 and 4: the overall classification performance is 81.69% (classification performance for digit 3: 85.94%; digit 4: 97.56%; digit 5: 59.42%). This result suggests that integration of newborn DGCs is beneficial for sequential learning of novel patterns.

In control 3, all DGCs keep plastic feedforward weights (learning rate $\eta > 0$) after pretraining and introduction of the novel digit 5, no matter if they became selective or not for the pretrained digits 3 and 4. We observe that in the case where all neurons are plastic, learning of the novel digit induces a change in selectivity of mature neurons. Several DGCs switch their selectivity to become sensitive to the novel digit (*Figure 3c*), while none of the previously unresponsive units becomes selective for presented patterns (compare with *Figure 1e*). In contrast to the model with neurogenesis, we observe a drop in classification performance to 90.92% (classification performance for digit 3: 85.45%; digit 4: 98.37%; digit 5: 88.90%). We find that the classification performance for digit 3 is the one which decreases the most. This is due to the fact that many DGCs previously selective for digit 3 modified their weights to become selective for digit 5. Importantly, the more novel patterns are introduced, the more overwriting of previously stored memories occurs. Hence, if all DGCs remain plastic, discrimination between a novel pattern and a familiar pattern stored long ago is impaired.

## Maturation of newborn neurons shapes the representation of novel patterns

Since each input pattern stimulates slightly different, yet overlapping, subsets of the 100 model DGCs in a sparse code such that about 20 DGCs respond to each pattern (*Figure 2g*), there is no simple one-to-one assignment between neurons and patterns. In order to visualize the activity patterns of the ensemble of DGCs, we perform dimensionality reduction. We construct a two-dimensional space using the activity patterns of the network at the end of the late phase of maturation of newborn DGCs trained with '3's, '4's and '5's. One axis connects the center of mass (in the 100-dimensional activity space) of all DGC responses to '3's with all responses to '5's (arbitrarily called 'axis 1') and the other axis those from '4's to '5's (arbitrarily called 'axis 2'). We then project the activity of the 100 model DGCs upon presentation of MNIST testing patterns onto those two axes, both at the end of the early and late phase of maturation of newborn DGCs (Materials and methods). Each two-dimensional projection is illustrated by a dot whose color corresponds to the digit class of the presented input pattern (blue for digit 3, green for digit 4, red for digit 5). Different input patterns within the same digit class cause different activation patterns of the DGCs, as depicted by extended clouds of dots of the same color (*Figure 4a,b*). Interestingly, an example pattern of a '5' that is visually similar to a '4' (characterized by the green cross) yields a DGC representation that lies closer to other '4's (green cloud of dots) than to typical '5's (red cloud of dots) (*Figure 4b*). Noteworthy the separation of the representation of '5's from '3's and '4's is better at end of the late phase (*Figure 4b*) when compared to the end of the early phase of maturation (*Figure 4a*). For instance, even though the pattern '5' corresponding to the orange cross is represented close to representations of '4's at the end of the early phase of maturation (green cloud of dots, *Figure 4a*), it is represented far from any '3's and '4's at the end of maturation (*Figure 4b*). The expansion of the representation of '5's into a previously empty subspace evolves as a function of time during the late phase of maturation (*Figure 4d*).

## Robustness of the model

Our results are robust to changes in network architecture. As mentioned earlier, neither the exact number of GABAergic neurons (*Supplementary file 2*), nor that of DGCs is critical. Indeed, a larger

network with 700 DGCs, thus mimicking the anatomically observed expansion factor of about 5 between EC and dentate gyrus (all other parameters unchanged), yields similar results (*Supplementary file 3*).

In the network with 700 DGCs, 275 cells remain unresponsive after pretraining with digits 3 and 4. In line with our earlier approach in the network with 100 DGCs, we can algoritmically replace all unresponsive neurons with newborn DGCs before patterns of digit 5 are added. Upon maturation, newborn DGC receptive fields provide a detailed representation of the prototypes of the novel digit 5 (*Figure 4—figure supplement 1*) and good classification performance is obtained (*Supplementary file 3*). Interestingly, due to the randomness of the recurrent connections, some newborn DGCs become selective for particular prototypes of the familiar (pretrained) digits 3 and 4 that are not already extensively represented by the network (see newborn DGCs selective for digit 4 highlighted by magenta squares in *Figure 4—figure supplement 1*).

As an alternative to replacing all unresponsive cells simultaneously, we can also replace only a fraction of them by newborn cells so as to simulate a continuous turn-over of cells. For example, if 119 of the 275 unresponsive cells are replaced by newborn DGCs before the start of presentations of digit 5, then these 119 cells become selective for different writing styles and generic features of the novel digit 5 (*Figure 4—figure supplement 2*) and allow a good classification performance of all three digits. On the other hand, replacing only 35 of the 275 unresponsive cells is not sufficient (*Supplementary file 3*). In an even bigger network with more than 144 EC cells and more than 700 DGCs, we could choose to replace 1% of the total DGC population per week by newborn cells, consistent with biology (*van Praag et al., 1999*; *Cameron and McKay, 2001*). Importantly, if only a small fraction of unresponsive cells are replaced at a given moment, other unresponsive cells remain available to be replaced later by newborn DGCs that are then ready to learn new stimuli.

Interestingly, the timing of the introduction of the novel stimulus is important. In our main neurogenesis model with 100 DGCs, we introduce the novel digit 5 at the beginning of the early phase of maturation, which consists of one epoch of MNIST training patterns (all patterns are presented once). If the novel digit is only introduced in the middle of the early phase (half epoch), it cannot be properly learned (classification performance for digit 5: 46.52%). However, if introduced after three-eights or one-quarter of the early phase, the novel digit can be picked out (classification performance for digit 5: 93.61% and 94.17%, respectively). We thus observe an increase in performance the earlier the novel digit is introduced after cell birth (classification performance for digit 5 was 95.18% when introduced at the beginning of the early phase of maturation). Therefore, our model predicts that a novel stimulus has to be introduced early enough with respect to newborn DGC maturation to be well discriminated and that the accuracy of discrimination is better the earlier it is introduced.

This could lead to an online scenario of our model, where adult-born DGCs are produced every day and different classes of novel patterns are introduced at different timepoints. To understand whether newborn DGCs in their early and late phase of maturation would interfere, two aspects should be kept in mind. First, since model newborn DGCs in the early phase of maturation do not project to other neurons yet, they do not influence the circuit and thus do not affect maturation of other newborn DGCs. Second, since model newborn DGCs in the late phase of maturation project to GABAergic neurons in the dentate gyrus, they will, just like mature cells, indirectly activate newborn DGCs that are in their early phase of maturation. As a result, early phase newborn DGCs will develop receptive fields that represent an average of all the stimuli that excite the mature and late phase newborn DGCs, which indirectly activate them. The ultimate selectivity of newborn DGCs is determined after the GABA-switch, when competition sets in, which makes those cells that have recently switched most sensitive to aspects of the input patterns that are not yet well represented by other cells. Therefore, in an online scenario, different model newborn DGCs would become selective for different novel patterns according to both their maturation stage with respect to presentation of the novel patterns, and the selectivity of mature and late phase newborn DGCs which indirectly activate them.

Finally, in our neurogenesis model, we have set the learning rate of mature DGCs to zero despite the observation that mature DGCs retain some plasticity (*Schmidt-Hieber et al., 2004*; *Ge et al., 2007*). We therefore studied a variant of the model in which mature DGCs also exhibit plasticity. First, we used our main model with 100 DGCs and 21 newborn DGCs. The implementation was identical, except that the learning rate of the mature DGCs was kept at a nonzero value during the

maturation of the 21 newborn DGCs. We do not observe a large change in classification performance, even if the learning rate of the mature cells is the same as that of newborn cells (*Supplementary file 4*). Second, we used our extended network with 700 DGCs to be able to investigate the effect of plastic mature DGCs while having a proportion of newborn cells matching experiments. We find that with 35 newborn DGCs (corresponding to the experimentally reported fraction of about 5%), plastic mature DGCs (with a learning rate half of that of newborn cells) improve classification performance (*Supplementary file 4*). This is due to the fact that several of the mature DGCs (that were previously selective for '3's or '4's) become selective for prototypes of the novel digit 5. Consequently, more than the 35 newborn DGCs specialize for digit 5, so that digit 5 is eventually represented better by the network with mature cell plasticity than the standard network where plasticity is limited to newborn cells. Note that those mature DGCs that had earlier specialized on writing styles of digit 3 or 4 similar to a digit 5 are most likely to retune their selectivity. If the novel inputs were very distinct from the pretrained familiar inputs, mature DGCs would be unlikely to develop selectivity for the novel inputs.

## Newborn DGCs become selective for similar novel patterns

To investigate whether our theory for integration of newborn DGCs can explain why adult dentate gyrus neurogenesis promotes discrimination of similar stimuli, but does not affect discrimination of distinct patterns (*Clelland et al., 2009*; *Sahay et al., 2011a*), we use a simplified competitive winner-take-all network (Materials and methods). It contains only as many DGCs as trained clusters, and the GABAergic inhibitory neurons are implicitly modeled through direct DGC-to-DGC inhibitory connections. DGCs are either silent or active (binary activity state, while in the detailed network DGCs had continuous firing rates). The synaptic plasticity rule is however the same as for the detailed network, with different parameter values (Materials and methods). We also construct an artificial data set (*Figure 5a,b*) that allows us to control the similarity $s$ of pairs of clusters (Materials and methods). The MNIST data set is not appropriate to distinguish similar from dissimilar patterns, because all digit clusters are similar and highly overlapping, reflected by a high within cluster dispersion (e.g. across the set of all '3') compared to the separation between clusters (e.g. typical '3' versus typical '5').

After a pretraining period, a first mature DGC responds to patterns of cluster 1 and a second mature DGC to those of cluster 2 (*Figure 5e,f*). We then fix the feedforward weights of those two DGCs and introduce a newborn DGC in the network. Thereafter, we present patterns from three clusters (the two pretrained ones, as well as a novel one), while the plastic feedforward weights of the newborn DGC are the only ones that are updated. We observe that the newborn DGC ultimately becomes selective for the novel cluster if it is similar ($s = 0.8$) to the two pretrained clusters (*Figure 5i*), but not if it is distinct ($s = 0.2$, *Figure 5j*). The selectivity develops in two phases. In the early phase of maturation of the newborn model cell, a pattern from the novel cluster that is similar to one of the pretrained clusters activates the mature DGC that has a receptive field closest to the novel pattern. The activated mature DGC drives the newborn DGC via lateral excitatory GABAergic connections to a firing rate where LTP is triggered at active synapses onto the newborn DGC. LTP also happens when a pattern from one of the pretrained clusters is presented. Thus, synaptic plasticity leads to a receptive field that reflects the average of all stimuli from all three clusters (*Figure 5g*).

To summarize our findings in a more mathematical language, we characterize the receptive field of the newborn cell by the vector of its feedforward weights. Analogous to the notion of a firing rate vector that represents the set of firing rates of an ensemble of neurons, the feedforward weight vector represents the set of weights of all synapses projecting onto a given neuron (*Figure 1b*). In the early phase of maturation, for similar clusters, the feedforward weight vector onto the newborn DGC grows in the direction of the center of mass of all three clusters (the two pretrained ones and the novel one), because for each pattern presentation, be it a novel pattern or a familiar one, one of the mature DGCs becomes active and stimulates the newborn cell (compare *Figure 5g* and *Figure 5k*). However, if the novel cluster has a low similarity to pretrained clusters, patterns from the novel cluster do not activate any of the mature DGCs. Therefore, the receptive field of the newborn cell reflects the average of stimuli from the two pretrained clusters only (compare *Figure 5h* and *Figure 5l*).

As a result of the different orientation of the feedforward weight vector onto the newborn DGC at the end of the early phase of maturation, two different situations arise in the late phase of

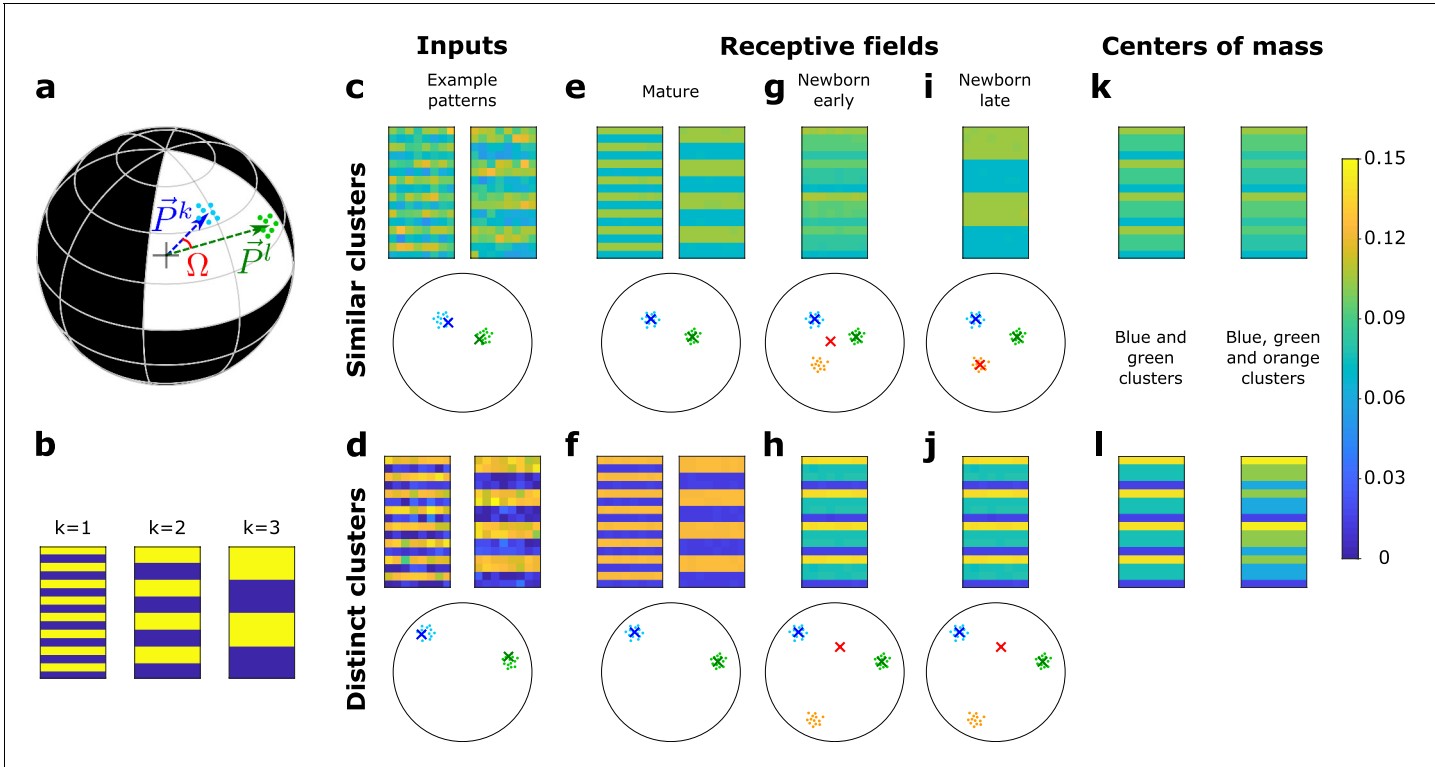

**Figure 5.** A newborn DGC becomes selective for similar but not distinct novel stimuli. (a) Center of mass of clusters $k$ and $l$ of an artificial data set ($\vec{P}_k$ and $\vec{P}_l$, respectively, separated by angle $\Omega$) are represented by arrows that point to the surface of a hypersphere. Dots represent individual patterns. (b) Center of mass of three clusters of the artificial data set, visualized as $16 \times 8$ pixel patterns. The two-dimensional arrangements and colors are for visualization only. (c, d) Example input patterns (activity of $16 \times 8$ input neurons) from clusters 1 and 2 for similar clusters (c, $s = 0.8$), and distinct clusters (d, $s = 0.2$). Below: dots correspond to patterns, crosses indicate the input patterns shown (schematic). (e, f) After pretraining with patterns from two clusters, the receptive fields (set of synaptic weights onto neurons 1 and 2) exhibit the center of mass of each cluster of input patterns (blue and green crosses). (g, h) Novel stimuli from cluster 3 (orange dots) are added. If the clusters are similar, the receptive field of the newborn DGC (red cross) moves toward the center of mass of the three clusters during its early phase of maturation (g), and if the clusters are distinct toward the center of mass of the two pretrained clusters (h). (i, j) Receptive field after the late phase of maturation for the case of similar (i) or distinct (j) clusters. (k, l) For comparison, the center of mass of all patterns of the blue and green clusters (left column) and of the blue, green, and orange clusters (right column) for the case of similar (k) or distinct (l) clusters. Color scale: input firing rate $\vec{x}$ or weight $\vec{w}_i$ normalized to $||\vec{w}_i|| = 1 = ||\vec{x}||$.

maturation, when lateral GABAergic connections are inhibitory. If the novel cluster is similar to the pretrained clusters, the weight vector onto the newborn DGC at the end of the early phase of maturation lies at the center of mass of all the patterns across the three clusters. Thus, it is closer to the novel cluster than the weight vector onto either of the mature DGCs (*Figure 5g*). So if a novel pattern is presented, the newborn DGC wins the competition between the three DGCs, and its feedforward weight vector moves toward the center of mass of the novel cluster (*Figure 5i*). By contrast, if the novel cluster is distinct, the weight vector onto the newborn DGC at the end of the early phase of maturation is located at the center of mass of the two pretrained clusters (*Figure 5h*). If a novel pattern is presented, no output unit is activated since their receptive fields are not similar enough to the input pattern. Therefore, the newborn DGC always stays silent and does not update its feedforward weights (*Figure 5j*). These results are consistent with studies that have suggested that dentate gyrus is only involved in the discrimination of similar stimuli, but not distinct stimuli (*Gilbert et al., 2001*; *Hunsaker and Kesner, 2008*). For discrimination of distinct stimuli, another pathway might be used, such as the direct EC to CA3 connection (*Yeckel and Berger, 1990*; *Fyhn et al., 2007*).

In conclusion, our model suggests that adult dentate gyrus neurogenesis promotes discrimination of similar patterns because newborn DGCs can ultimately become selective for novel stimuli, which are similar to already learned stimuli. On the other hand, newborn DGCs fail to represent novel distinct stimuli, precisely because they are too distinct from other stimuli already represented by the

network. Presentation of novel distinct stimuli in the late phase of maturation therefore does not induce synaptic plasticity of the newborn DGC feedforward weight vector toward the novel stimuli. In the simplified network, the transition between similar and distinct can be determined analytically (Materials and methods). This analysis clarifies the importance of the switch from cooperative dynamics (excitatory interactions) in the early phase to competitive dynamics (inhibitory interactions) in the late phase of maturation.

## Upon successful integration the receptive field of a newborn DGC represents an average of novel stimuli

With the simplified model network, it is possible to analytically compute the maximal strength of the DGC receptive field via the L2-norm of the feedforward weight vector onto the newborn DGC (Materials and methods). In addition, the angle between the center of mass of the novel patterns and the feedforward weight vector onto the adult-born DGC can also be analytically computed (Materials and methods). To illustrate the analytical results and characterize the evolution of the receptive field of the newborn DGC, we thus examine the angle φ of the feedforward weight vector with the center of mass of the novel cluster (i.e. the average of the novel stimuli), as a function of maturation time (*Figure 6b,c*, *Figure 6—figure supplement 1*).

In the early phase of maturation, the feedforward weight vector onto the newborn DGC grows, while its angle with the center of mass of the novel cluster stays constant (*Figure 6—figure supplement 1*). In the late phase of maturation, the angle φ between the center of mass of the novel cluster and the feedforward weight vector onto the newborn DGC decreases in the case of similar patterns (*Figure 6c*, *Figure 6—figure supplement 1*), but not in the case of distinct patterns (*Figure 6—figure supplement 1*), indicating that the newborn DGC becomes selective for the novel cluster for similar but not for distinct patterns.

The analysis of the simplified model thus leads to a geometric picture that helps us to understand how the similarity of patterns influences the evolution of the receptive field of the newborn DGC before and after the switch from excitation to inhibition of the GABAergic input. For novel patterns that are similar to known patterns, the receptive field of a newborn DGC at the end of maturation represents the average of novel stimuli.

## The cooperative phase of maturation promotes pattern separation for any dimensionality of input data

Despite the fact that input patterns in our model represent the activity of 144 or 128 model EC cells, the effective dimensionality of the input data was significantly below 100 because the clusters for different input classes were rather concentrated around their respective center of mass. We define the effective input dimensionality as the participation ratio (*Mazzucato et al., 2016*; *Litwin-Kumar et al., 2017*) (Materials and methods). Using this definition, the input data of both the MNIST $12 \times 12$ patterns from digits 3, 4, and 5 and the seven clusters of the handmade dataset for similar patterns ($s = 0.8$) are relatively low-dimensional ($PR = 19$ of a maximum of 144, and $PR = 11$ of a maximum of 128, respectively). We emphasize that in both cases the spread of the input data around the cluster center implies that the effective dimensionality is larger than the number of clusters. In natural settings, we expect the input data to have even higher dimension. Therefore, here we investigate the effect of dimensionality of the input data on our neurogenesis model by increasing the spread around the cluster centers.

We use our simplified network model and create similar artificial datasets ($s = 0.8$) with different values for the concentration parameter $\kappa$ (Materials and methods). The smaller the $\kappa$, the broader the distributions around their center of mass; hence, the larger the overlap of patterns generated from different cluster distributions. Therefore, we can increase the effective dimensionality of the input by decreasing the concentration parameter $\kappa$. First, as expected from our analytical analysis (Materials and methods), we find that the broader the cluster distributions the smaller the length of the feedforward weight vector onto newborn DGCs (from just below 1.5 with $\kappa = 10^4$ to about 1.35 with $\kappa = 6 \cdot 10^2$). Second, we examine the ability of the simplified network to discriminate input patterns coming from input spaces with different dimensionalities. To do so, we compare our neurogenesis model (Neuro.) with a random initialization model (RandInitL.). In both cases, two DGCs are pretrained with patterns from two clusters, as above. Then we fix the weights of the two mature

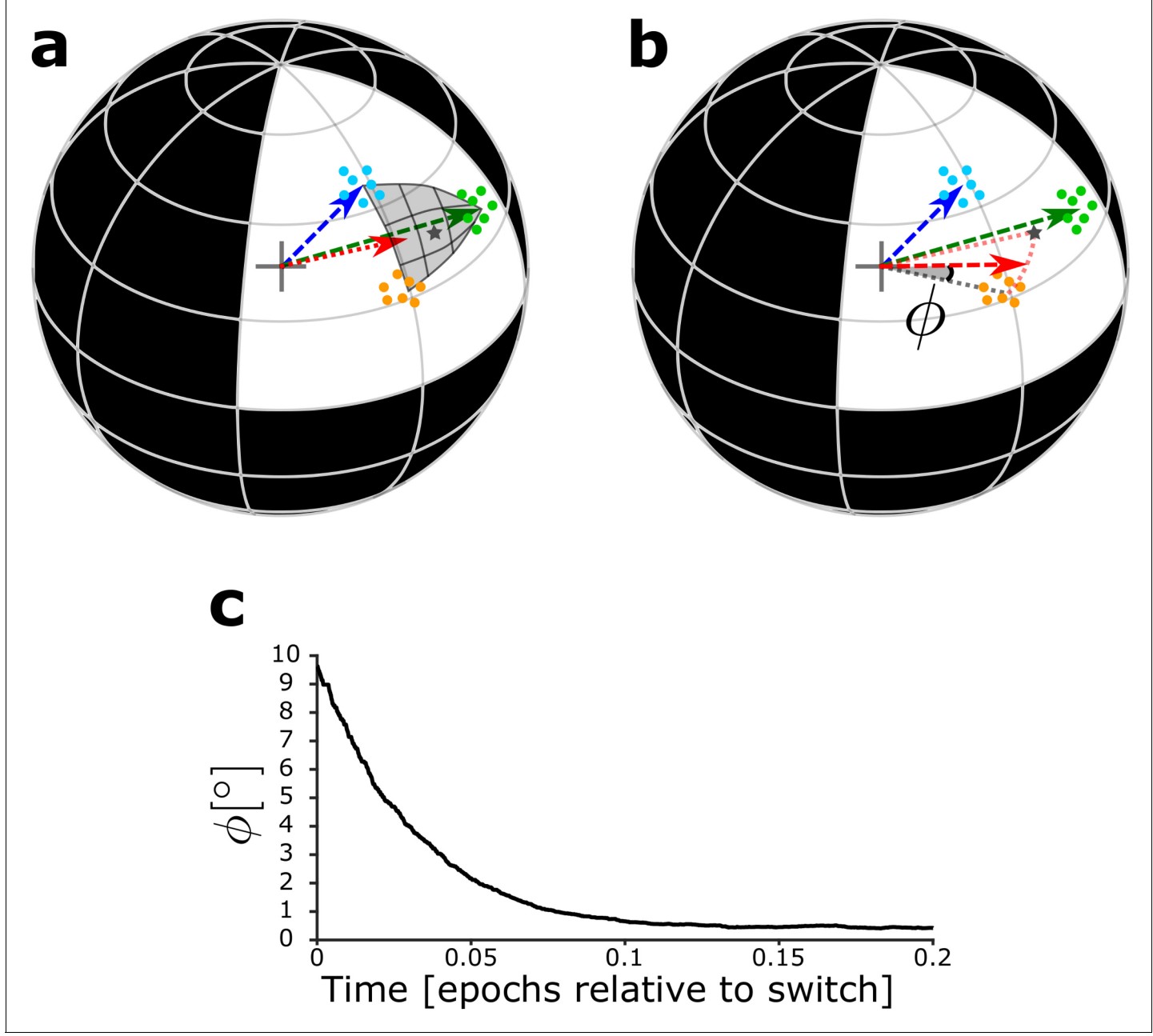

**Figure 6.** Maturation dynamics for similar patterns. (**a**) Schematics of the unit hypersphere with three clusters of patterns (colored dots) and three scaled feedforward weight vectors (colored arrows). After pretraining, the blue and green weight vectors point to the center of mass of the corresponding clusters. Patterns from the novel cluster (orange points) are presented only later to the network. During the early phase of maturation, the newborn DGC grows its vector of feedforward weights (red arrow) in the direction of the subspace of patterns which indirectly activate the newborn cell (dark grey star: center of mass of the presented patterns, located below the part of the sphere surface highlighted in grey). (**b**) During the late phase of maturation, the red vector turns toward the novel cluster. The symbol φ indicates the angle between the center of mass of the novel cluster and the feedforward weight vector onto the newborn cell. (**c**) The angle φ decreases in the late phase of maturation of the newborn DGC if the novel cluster is similar to the previously stored clusters. Its final average value of $\phi \approx 0.4°$ is caused by the jitter of the weight vector around the center of mass of the novel cluster.

The online version of this article includes the following figure supplement(s) for figure 6:

**Figure supplement 1.** Evolution of the feedforward weight vector onto the newborn DGC.

DGCs and introduce patterns from a third cluster as well as a newborn DGC. For the neurogenesis case, after maturation of the newborn DGC we fix its weights (while for the random initialization model we keep them plastic) upon introduction of patterns from a fourth cluster as well as another newborn DGC, and so on until the network contains seven DGCs and patterns from the full dataset of seven clusters have been presented. We compare our neurogenesis model, where each newborn DGC starts with zero weights and undergo a two-phase maturation (one epoch per phase), with a random initialization model where each newborn DGC is directly fully integrated into the circuit and whose feedforward weight vector is randomly initialized with a length of 0.1 (RandInitL.) and is then learned for two epochs.

Since clusters can be highly overlapping, we assess discrimination performance by computing the reconstruction error at the end of training. Reconstruction error is evaluated analogously to classification error, except that the readout layer has the task of an autoencoder: it contains as many readout units as there are input units. Reconstruction error is the mean squared distance between the input vector and the reconstructed output vector based on testing patterns. We observe that for any dimensionality of the input space, even as high as 97-dimensional, the neurogenesis model performs better (has a lower total reconstruction error) than the random initialization model (*Supplementary file 5*). Indeed, in the neurogenesis case newborn DGCs grow their feedforward weights (from zero) in the direction of presented input patterns in their early cooperative phase of maturation and can later become selective for novel patterns during the competitive phase. In contrast, since the random initialization model has no early cooperative phase, the newborn DGC weight vector does not grow unless an input pattern is by chance well aligned with its randomly initialized weight vector (which is unlikely in a high-dimensional input space). We get similar results for a larger initialization of the synaptic weights (e.g. the length of the weight vector at birth is set to 1, results not shown). Importantly, in high input dimensions, the advantage of a larger weight vector length at birth in the random initialization model is overridden by the capability of newborn DGCs to grow their weight vector in the appropriate direction during their early cooperative phase of maturation. Finally, we note that even if the length of the feedforward weight vector onto newborn DGCs is set to 1.5 (RandInitH., *Supplementary file 5*), which is the upper bound according to our analytical results (Materials and methods), the random initialization model performs worse than the neurogenesis model for low up to relatively high-dimensional input spaces ($PR = 83$, *Supplementary file 5*) despite its advantage in the competition conferred by the longer weight vector. It is only when input clusters are extremely broad and overlapping that the random initialization model performs similarly to the neurogenesis model ($PR = 90, 97$, *Supplementary file 5*). In other words, a random initialization at full length of weight vectors works well if input data is homogeneously distributed on the positive quadrant of the unit sphere but fails if the input data is clustered in a few directions. Moreover, random initialization requires that synaptic weights are large from the start which is biologically not plausible. In summary, the two-phase neurogenesis model is advantageous because the feedforward weights onto newborn cells can start at arbitrarily small values; their growth is, during the cooperative phase, guided to occur in a direction that is relevant for the task at hand; the final competitive phase eventually enables specialization onto novel inputs.

## Discussion

While experimental studies, such as manipulating the ratio of NKCC1 to KCC2, suggest that the switch from excitation to inhibition of the GABAergic input onto adult-born DGCs is crucial for their integration into the preexisting circuit (*Ge et al., 2006*; *Alvarez et al., 2016*) and that adult dentate gyrus neurogenesis promotes pattern separation (*Clelland et al., 2009*; *Sahay et al., 2011a*; *Jessberger et al., 2009*), the link between channel properties and behavior has remained puzzling (*Sahay et al., 2011b*; *Aimone et al., 2011*). Our modeling work shows that the GABA-switch enables newborn DGCs to become selective for novel stimuli, which are similar to familiar, already-stored, representations, consistent with the experimentally observed function of pattern separation (*Clelland et al., 2009*; *Sahay et al., 2011a*; *Jessberger et al., 2009*).

Previous modeling studies already suggested that newborn DGCs integrate novel inputs into the representation in dentate gyrus (*Chambers et al., 2004*; *Becker, 2005*; *Crick and Miranker, 2006*; *Wiskott et al., 2006*; *Chambers and Conroy, 2007*; *Appleby and Wiskott, 2009*; *Aimone et al., 2009*; *Weisz and Argibay, 2009*; *Temprana et al., 2015*; *Finnegan and Becker, 2015*;

*DeCostanzo et al., 2018*). However, our work differs from them in four important aspects. First of all, we implement an unsupervised biologically plausible plasticity rule, while many studies used supervised algorithmic learning rules (*Chambers et al., 2004*; *Becker, 2005*; *Chambers and Conroy, 2007*; *Weisz and Argibay, 2009*; *Finnegan and Becker, 2015*; *DeCostanzo et al., 2018*). Second, as we model the formerly neglected GABA-switch, the connection weights from EC to newborn DGCs are grown from small values through cooperativity in the early phase of maturation. This integration step was mostly bypassed in earlier models by initialization of the connectivity weights toward newborn DGCs to random, yet fully grown values (*Crick and Miranker, 2006*; *Aimone et al., 2009*; *Weisz and Argibay, 2009*; *Finnegan and Becker, 2015*). Third, as the dentate gyrus network is commonly modeled as a competitive network, weight normalization is crucial. In our framework, competition occurs during the late phase of maturation. Previous modeling works either applied algorithmic weight normalization or hard bounds on the weights at each iteration step (*Crick and Miranker, 2006*; *Aimone et al., 2009*; *Weisz and Argibay, 2009*; *Temprana et al., 2015*; *Finnegan and Becker, 2015*). Instead, our plasticity rule includes heterosynaptic plasticity, which intrinsically softly bounds connectivity weights by a homeostatic effect. Finally, although some earlier computational models of adult dentate gyrus neurogenesis could explain the pattern separation abilities of newborn cells, separation was obtained independently of the similarity between the stimuli. Contrarily to experimental data, no distinction was made between similar and distinct patterns (*Chambers et al., 2004*; *Becker, 2005*; *Crick and Miranker, 2006*; *Wiskott et al., 2006*; *Chambers and Conroy, 2007*; *Aimone et al., 2009*; *Appleby and Wiskott, 2009*; *Weisz and Argibay, 2012*; *Temprana et al., 2015*; *Finnegan and Becker, 2015*; *DeCostanzo et al., 2018*). To our knowledge, we present the first model that can explain both (1) how adult-born DGCs integrate into the preexisting network and (2) why they promote pattern separation of similar stimuli and not distinct stimuli.

Our work emphasizes why a two-phase maturation of newborn DGCs is beneficial for proper integration in the preexisting network. From a computational perspective, the early phase of maturation, when GABAergic inputs onto newborn DGCs are excitatory, corresponds to cooperative unsupervised learning. Therefore, the synapses grow in the direction of patterns that indirectly activate the newborn DGCs via GABAergic interneurons (*Figure 6a*). At the end of the early phase of maturation, the receptive field of a newborn DGC represents the center of mass of all input patterns that led to its (indirect) activation. In the late phase of maturation, GABAergic inputs onto newborn DGCs become inhibitory, so that lateral interactions change from cooperation to competition, causing a shift of the receptive fields of the newborn DGCs toward novel features (*Figure 6b*). At the end of maturation, newborn DGCs are thus selective for novel inputs. This integration mechanism is in agreement with the experimental observation that newborn DGCs are broadly tuned early in maturation, yet highly selective at the end of maturation (*Marín-Burgin et al., 2012*; *Danielson et al., 2016*). Loosely speaking, the cooperative phase of excitatory GABAergic input promotes the growth of the synaptic weights coarsely in the relevant direction, whereas the competitive phase of inhibitory GABAergic input helps to specialize on detailed, but potentially important differences between patterns.

In the context of theories of unsupervised learning, the switch of lateral GABAergic input to newborn DGCs from excitatory to inhibitory provides a biological solution to the 'problem of unresponsive units' (*Hertz et al., 1991*). Unsupervised competitive learning has been used to perform clustering of input patterns into a few categories (*Rumelhart and Zipser, 1985*; *Grossberg, 1987*; *Kohonen, 1989*; *Hertz et al., 1991*; *Du, 2010*). Ideally, after learning of the feedforward weights between an input layer and a competitive network, input patterns that are distinct from each other activate different neuron assemblies of the competitive network. After convergence of competitive Hebbian learning, the vector of feedforward weights onto a given neuron points to the center of mass of the cluster of input patterns for which it is selective (*Kohonen, 1989*; *Hertz et al., 1991*). Yet, if the synaptic weights are randomly initialized, it is possible that the set of feedforward weights onto some neurons of the competitive network point in a direction 'quasi-orthogonal' (Materials and methods) to the subspace of the presented input patterns. Therefore, those neurons, called 'unresponsive units', will never get active during pattern presentation. Different learning strategies have been developed in the field of artificial neural networks to avoid this problem (*Grossberg, 1976*; *Bienenstock et al., 1982*; *Rumelhart and Zipser, 1985*; *Grossberg, 1987*; *DeSieno, 1988*; *Kohonen, 1989*; *Hertz et al., 1991*; *Du, 2010*). However, most of these algorithmic approaches lack a

biological interpretation. In our model, weak synapses onto newborn DGCs form spontaneously after neuronal birth. The excitatory GABAergic input in the early phase of maturation drives the growth of the synaptic weights in the direction of the subspace of presented patterns that succeed in activating some of the mature DGCs. Hence, the early cooperative phase of maturation can be seen as a smart initialization of the synaptic weights onto newborn DGCs, close enough to novel patterns so as to become selective for them in the late competitive phase of maturation. However, the cooperative phase is helpful only if the novel patterns are similar to the input statistics defined by the set of known (familiar) patterns.

Our results are in line with the classic view that dentate gyrus is responsible for decorrelation of inputs (*Marr, 1969*; *Albus, 1971*; *Marr, 1971*; *Rolls and Treves, 1998*), a necessary step for differential storage of similar memories in CA3, and with the observation that dentate gyrus lesions impair discrimination of similar but not distinct stimuli (*Gilbert et al., 2001*; *Hunsaker and Kesner, 2008*). To discriminate distinct stimuli, another pathway might be involved, such as the direct EC to CA3 connection (*Yeckel and Berger, 1990*; *Fyhn et al., 2007*).

The parallel of neurogenesis in dentate gyrus and olfactory bulb suggests that similar mechanisms could be at work in both areas. Yet, even though adult olfactory bulb neurogenesis seems to have a similar functional role to adult dentate gyrus neurogenesis (*Sahay et al., 2011b*), follow a similar integration sequence and undergo a GABA-switch from excitatory to inhibitory, the circuits are different in several aspects. First, while newborn neurons in dentate gyrus are excitatory, newborn cells in the olfactory bulb are inhibitory. Second, the newborn olfactory cells start firing action potentials only once they are well integrated (*Carleton et al., 2003*). Therefore, in view of a transfer of results to the olfactory bulb, it would be interesting to adjust our model of adult dentate gyrus neurogenesis accordingly. For example, a voltage-based synaptic plasticity rule could be used to account for subthreshold plasticity mechanisms (*Clopath et al., 2010*).

Our model of transition from an early cooperative phase to a late competitive phase makes specific predictions, at the behavioral and cellular level. In our model, the early cooperative phase of maturation can only drive the growth of synaptic weights onto newborn cells if they are indirectly activated by mature DGCs through GABAergic input, which has an excitatory effect due to the high NKCC1/KCC2 ratio early in maturation. Therefore, our model predicts that NKCC1-knockout mice would be impaired in discriminating similar contexts or objects because newborn cells stay silent due to lack of indirect activation. The feedforward weight vector onto newborn DGCs could not grow in the early phase and newborn DGCs could not become selective for novel inputs. Therefore, our model predicts that since newborn DGCs are poorly integrated into the preexisting circuit, they are unlikely to survive. If, however, in the same paradigm newborn cells are activated by light-induced or electrical stimulation, we predict that they become selective to novel patterns. Thus discrimination abilities would be restored and newborn DGCs are likely to survive. Analogously, we predict that using inducible NKCC1-knockout mice, animals would gradually be impaired in discrimination tasks after induced knockout and reach a stable maximum impairment about 3 weeks after the start of induced knockout.

Experimental observations support the importance of the switch from early excitation to late inhibition of the GABAergic input onto newborn DGCs. An absence of early excitation using NKCC1-knockout mice has been shown to strongly affect synapse formation and dendritic development in vivo (*Ge et al., 2006*). Conversely, a reduction in inhibition in the dentate gyrus through decrease in KCC2 expression has been associated with epileptic activity (*Pathak et al., 2007*; *Barmashenko et al., 2011*). An analogous switch of the GABAergic input has been observed during development, and its proper timing has been shown to be crucial for sensorimotor gating and cognition (*Wang and Kriegstein, 2011*; *Furukawa et al., 2017*). In addition to early excitation and late inhibition, our theory also critically depends on the time scale of the switching process. In our model, the switch makes an instantaneous transition between early and late phase of maturation. Several experimental results have suggested that the switch is indeed sharp and occurs within a single day, both during development (*Khazipov et al., 2004*; *Tyzio et al., 2007*; *Leonzino et al., 2016*) and adult dentate gyrus neurogenesis (*Heigele et al., 2016*). Furthermore, in hippocampal cell cultures, expression of KCC2 is upregulated by GABAergic activity but not affected by glutamatergic activity (*Ganguly et al., 2001*). A similar process during adult dentate gyrus neurogenesis would increase the number of newborn DGCs available for representing novel features by advancing the timing of their switch. In this way, instead of a few thousands of newborn DGCs ready to switch (3–6% of the

whole population [*van Praag et al., 1999*; *Cameron and McKay, 2001*], divided by 30 days), a larger fraction of newborn DGCs would be made available for coding, if appropriate stimulation occurs. Finally, while neurotransmitter switching has been observed following sustained stimulation for hours to days (*Li et al., 2020*), it is still unclear if it has the same functional role as the GABA-switch in our model. In particular, it remains an open question if neurotransmitter switching promotes the integration of neurons in the same way as our model GABA-switch does in the context of adult dentate gyrus neurogenesis.

To conclude, our theory for integration of adult-born DGCs suggests that newborn cells have a coding – rather than a modulatory – role during dentate gyrus pattern separation function. Our theory highlights the importance of GABAergic input in adult dentate gyrus neurogenesis and links the switch from excitation to inhibition to the integration of newborn DGCs into the preexisting circuit. Finally, it illustrates how Hebbian plasticity of EC to DGC synapses along with the switch make newborn cells suitable to promote pattern separation of similar but not distinct stimuli, a long-standing mystery in the field of adult dentate gyrus neurogenesis (*Sahay et al., 2011b*; *Aimone et al., 2011*).

## Materials and methods

### Network architecture and neuronal dynamics

DGCs are the principal cells of the dentate gyrus. They mainly receive excitatory projections from the EC through the perforant path and GABAergic inputs from local interneurons, as well as excitatory input from Mossy cells. They project to CA3 pyramidal cells and inhibitory neurons, as well as local Mossy cells (*Acsády et al., 1998*; *Henze et al., 2002*; *Amaral et al., 2007*; *Temprana et al., 2015*; *Figure 1—figure supplement 1*). In our model, we omit Mossy cells for simplicity and describe the dentate gyrus as a competitive circuit consisting of $N_{DGC}$ DGCs and $N_I$ GABAergic interneurons (*Figure 1b*). The activity of $N_{EC}$ neurons in EC represents an input pattern $\vec{x} = (x_1, x_2, ..., x_{N_{EC}})$. Because the perforant path also induces strong feedforward inhibition in the dentate gyrus (*Li et al., 2013*), we assume that the effective EC activity is normalized, such that $||\vec{x}|| = 1$ for any input pattern $\vec{x}$ (*Figure 1—figure supplement 1*). We use $P$ different input patterns $\vec{x}^\mu$, $1 \leqslant \mu \leqslant P$ in the simulations of the model.

In our network, model EC neurons have excitatory all-to-all connections to the DGCs. In rodent hippocampus, spiking mature DGCs activate interneurons in dentate gyrus, which in turn inhibit other mature DGCs (*Temprana et al., 2015*; *Alvarez et al., 2016*). In our model, the DGCs are thus recurrently connected with inhibitory neurons (*Figure 1b*). Connections from DGCs to interneurons exist in our model with probability $p_{IE}$ and have a weight $w_{IE}$. Similarly, connections from interneurons to DGCs occur with probability $p_{EI}$ and have a weight $w_{EI}$. All parameters are reported in *Table 1* (Biologically plausible network).

Before an input pattern is presented, all rates of model DGCs are initialized to zero. We assume that the DGCs have a frequency–current curve that is given by a rectified hyperbolic tangent (*Dayan and Abbott, 2001*), which is similar to the frequency–current curve of spiking neuron models with refractoriness (*Gerstner et al., 2014*). Moreover, we exploit the equivalence of two common firing rate equations (*Miller and Fumarola, 2012*) and let the firing rate $\nu_i$ of DGC $i$ upon stimulation with input pattern $\vec{x}$ evolve according to:

$$\tau_m \frac{\mathrm{d}\nu_i}{\mathrm{d}t} = -\nu_i + \tanh\left(\frac{[I_i - b_i]_+}{L}\right) \tag{2}$$

where $[.]_+$ denotes rectification: $[a]_+ = a$ for $a > 0$ and zero otherwise. Here, $b_i$ is a firing threshold, $L$ is the smoothness parameter of the frequency–current curve ($L^{-1}$ is the slope of the frequency–current curve at the firing threshold), and $I_i$ the total input to cell $i$:

$$I_i = \sum_{j=1}^{N_{EC}} w_{ij} x_j + \sum_{k=1}^{N_I} w_{ik}^{EI} \nu_k^I \tag{3}$$

with $x_j$ the activity of EC input neuron $j$, $w_{ij} \geqslant 0$ the feedforward weight from EC input neuron $j$ to DGC $i$, and $w_{ik}^{EI}$ the weight from inhibitory neuron $k$ to DGC $i$. The sum runs over all inhibitory neurons, but the weights are set to $w_{ik}^{EI} = 0$ if the connection is absent. The firing rate $\nu_i$ is unit-free and

normalized to a maximum of 1, which we interpret as a firing rate of 10 Hz. We take the synaptic weights as unit-less parameters such that $I_i$ is also unit-free.

The firing rate $\nu_k^I$ of inhibitory neuron $k$, is defined as:

$$\tau_{\text{inh}}\frac{\mathrm{d}\nu_k^I}{\mathrm{d}t} = -\nu_k^I + [I_k^I - p^* N_{DGC}]_+ \tag{4}$$

with $p^*$ a parameter which relates to the desired ensemble sparsity, and $I_k^I$ the total input toward interneuron $k$, given as:

$$I_k^I = \sum_{i=1}^{N_{DGC}} w_{ki}^{IE} \nu_i \tag{5}$$

with $w_{ki}^{IE}$ the weight from DGC $i$ to inhibitory neuron $k$. (We set $w_{ki}^{IE} = 0$ if the connection is absent.) The feedback from inhibitory neurons ensures a sparse activity of model DGCs for each pattern. With $p^* = 0.1$ we find that more than 70% of model DGCs are silent (firing rate < 1 Hz [**Senzai and Buzsáki, 2017**]) when an input pattern is presented, and less than 10% are highly active (firing rate > 1 Hz) (**Figure 2c,d**), consistent with the experimentally observed sparse activity in dentate gyrus (**Chawla et al., 2005**).

## Plasticity rule

Projections from EC onto newborn DGCs exhibit Hebbian plasticity (**Schmidt-Hieber et al., 2004**; **Ge et al., 2007**; **McHugh et al., 2007**). Therefore, in our model, the connections from EC neurons to DGCs are plastic, following a Hebbian learning rule that exhibits LTD or LTP depending on the firing rate $\nu_i$ of the postsynaptic cell (**Bienenstock et al., 1982**; **Artola et al., 1990**; **Sjöström et al., 2001**; **Pfister and Gerstner, 2006**). Input patterns, $\vec{x}^\mu$, $1 \leqslant \mu \leqslant P$, are presented in random order. For each input pattern, we let the firing rate converge for a time $T$ where $T$ was chosen long enough to achieve convergence to a precision of $10^{-6}$. After $n-1$ presentations (i.e. at time $(n-1)\cdot T$), the weight vector has value $w_{ij}^{(n-1)}$. We then present the next pattern and update at time $n\cdot T$ ($w_{ij}^{(n)} = w_{ij}^{(n-1)} + \Delta w_{ij}$), according to the following plasticity rule (**Equation (1)**, written here for convenience):

$$\Delta w_{ij} = \eta\left\{ \gamma x_j \nu_i [\nu_i - \theta]_+ - \alpha x_j \nu_i [\theta - \nu_i]_+ - \beta w_{ij} [\nu_i - \theta]_+ \nu_i^3 \right\}$$

where $x_j$ is the firing rate of presynaptic EC input neuron $j$, $\nu_i$ the firing rate of postsynaptic DGC $i$, $\eta$ the learning rate, $\theta$ marks the transition from LTD to LTP, and the relative strength $\alpha$, $\gamma$ of LTP and LTD depend on $\theta$ via $\alpha = \frac{\alpha_0}{\theta^3} > 0$ and $\gamma = \gamma_0 - \theta > 0$. The values of the parameters $\alpha_0$, $\gamma_0$, $\beta$, and $\theta$ are given in **Table 1** (Biologically plausible network). The weights are hard-bounded from below at 0, i. e., if **Equation (1)** leads to a new weight smaller than zero, $w_{ij}$ is set to zero. The first two terms of **Equation (1)** are a variation of the BCM rule (**Bienenstock et al., 1982**). The third term implements heterosynaptic plasticity (**Chistiakova et al., 2014**; **Zenke and Gerstner, 2017**) with three important features: first, heterosynaptic plasticity has a negative sign and therefore leads to synaptic depression; second, heterosynaptic plasticity sets in above a threshold ($\nu_i > \theta$) that is the same threshold as that for LTP, so that if LTP occurs at some synapses LTD is induced at other synapses; third, above threshold the dependence upon the postsynaptic firing rate $\nu_i$ is supra-linear. The interaction of the three different terms in the plasticity rule has several consequences. Because the first two terms of the plasticity rule are Hebbian ('homosynaptic') and proportional to the presynaptic activity $x_j$, the active DGCs ($\nu_i > \theta$) update their feedforward weights in direction of the input pattern $\vec{x}$. Moreover, whenever LTP occurs at some synapses, all weights onto neuron $i$ are downregulated heterosynaptically by an amount that increases supra-linearly with the postsynaptic rate $\nu_i$, implicitly controlling the length of the weight vector (see below) similar to synaptic homeostasis (**Turrigiano et al., 1998**) but on a rapid time scale (**Zenke and Gerstner, 2017**). Analogous to learning in a competitive network (**Kohonen, 1989**; **Hertz et al., 1991**), the vector of feedforward weights onto active DGCs will move toward the center of mass of the cluster of patterns they are selective for, as we will discuss now.

For a given input pattern $\vec{x}^\mu$, there are three fixed points for the postsynaptic firing rate: $\nu_i = 0$, $\nu_i = \theta$, and $\nu_i = \hat{\nu}_i$ (the negative root is omitted because $\nu_i \geqslant 0$ due to *Equation (2)*). For $\nu_i < \theta$, there is LTD, so the weights move toward zero: $w_{ij} \rightarrow 0$, while for $\nu_i > \theta$, there is LTP, so the weights move toward $w_{ij} \rightarrow \frac{\gamma x_j^\mu}{\beta \hat{\nu}_i^2}$ (*Figure 1c*). The value of $\hat{\nu}_i$ is defined implicitly by the network *Equations (2–5)*. If a pattern $\vec{x}^\mu$ is presented only for a short time these fixed points are not reached during a single pattern presentation.

## Winners, losers, and quasi-orthogonal inputs

We define the winners as the DGCs that become strongly active ($\nu_i > \theta$) during presentation of an input pattern. Since the input patterns are normalized to have an L2-norm of 1 ($||\vec{x}^\mu|| = 1$ by construction), and the L2-norm of the feedforward weight vectors is bounded (see Section Direction and length of the weight vector), the winning units are the ones whose weight vectors $\vec{w}_i$ (row of the feedforward connectivity matrix) align best with the current input pattern $\vec{x}^\mu$.

We emphasize that all synaptic weights and all presynaptic firing rates $\nu_j$ are non-negative: $w_{ij} \geqslant 0$ and $\nu_j \geqslant 0$. Thus, both the weight vectors and the vectors of input firing rates live in the positive quadrant. The angle between an input pattern $\vec{x}^\mu$ and the weight vector $\vec{w}_i$ of neuron $i$ can be at most ninety degrees. We say that an input pattern $\vec{x}^\mu$ is 'quasi-orthogonal' to a weight vector $\vec{w}_i$ if, in the stationary state, the input is not sufficient to activate neuron $i$, i.e., $I_i = \sum_{j=1}^{N_{EC}} w_{ij} x_j + \sum_{k=1}^{N_I} w_{ik}^{EI} \nu_k^I < b_i$. If an input pattern $\vec{x}^\mu$ is quasi-orthogonal to a weight vector $\vec{w}_i$, then neuron $i$ does not fire in response to $\vec{x}^\mu$ after the stimulus has been applied for a long enough time. Note that for a case without inhibitory neurons and with $b_i \rightarrow 0$, we recover the standard orthogonality condition, but for finite $b_i > 0$ quasi-orthogonality corresponds to angles larger than some reference angle.

## Direction and length of the weight vector

Let us denote the ensemble of patterns for which neuron $i$ is a winner by $C_i$ and call this the set of winning patterns ($C_i = \{\mu | \nu_i > \theta\}$). Suppose that neuron $i$ is quasi-orthogonal to all other patterns, so that for all $\mu \notin C_i$, we have $\nu_i = 0$. Then the feedforward weight vector of neuron $i$ converges in expectation to:

$$\vec{w}_i = \frac{\gamma}{\beta} \frac{\langle G_1(\nu_i) \vec{x} \rangle_{\mu \in C_i}}{\langle G_2(\nu_i) \rangle_{\mu \in C_i}} \tag{6}$$

where $G_1(\nu_i) = (\nu_i - \theta) \nu_i$ and $G_2(\nu_i) = (\nu_i - \theta) \nu_i^3$. Hence $\vec{w}_i$ is a weighted average over all winning patterns.

The squared length of the feedforward weight vector can be computed by multiplying *Equation (6)* with $\vec{w}_i$:

$$||\vec{w}_i||^2 = \vec{w}_i \cdot \vec{w}_i = \frac{\gamma}{\beta} \frac{\langle G_1(\nu_i)(\vec{w}_i \cdot \vec{x}) \rangle_{\mu \in C_i}}{\langle G_2(\nu_i) \rangle_{\mu \in C_i}} \tag{7}$$

Since input patterns have length one, the scalar product on the right-hand side can be rewritten as $\vec{w}_i \cdot \vec{x} = ||\vec{w}_i|| \cos(\alpha)$ where $\alpha$ is the angle between the weight vector and pattern $\vec{x}$. Division by $||\vec{w}_i||$ yields the L2-norm of the feedforward weight vector:

$$||\vec{w}_i|| = \frac{\gamma}{\beta} \frac{\langle G_1(\nu_i) \cos(\alpha) \rangle_{\mu \in C_i}}{\langle G_2(\nu_i) \rangle_{\mu \in C_i}} \tag{8}$$

where the averages run, as before, over all winning patterns.

Let us now derive bounds for $||\vec{w}_i||$. First, since $\cos(\alpha) \leqslant 1$ we have $\langle G_1(\nu_i) \cos(\alpha) \rangle_{\mu \in C_i} \leqslant \langle G_1(\nu_i) \rangle_{\mu \in C_i}$. Second, since for all winning patterns $\nu_i > \theta$, where $\theta$ is the LTP threshold, we have $\langle G_2(\nu_i) \rangle_{\mu \in C_i} \geqslant \langle (\nu_i - \theta) \nu_i \rangle \theta^2$. Thus the length of the weight vector is finite and bounded by:

$$||\vec{w}_i|| \leqslant \frac{\gamma}{\beta} \frac{\langle G_1(\nu_i) \rangle_{\mu \in C_i}}{\langle G_2(\nu_i) \rangle_{\mu \in C_i}} \leqslant \frac{\gamma}{\beta} \frac{1}{\theta^2} \tag{9}$$

It is possible to make the second bound tighter if we find the winning pattern with the smallest firing rate $\nu_{\min}$ such that $\nu_i \geqslant \nu_{\min} \; \forall i \in C_i$:

$$||\vec{w}_i|| \leqslant \frac{\gamma}{\beta} \frac{1}{(\nu_{\min})^2} \tag{10}$$

The bound is reached if neuron $i$ is winner for a single input pattern.

We can also derive a lower bound. For a pattern $\mu \in C_i$, let us write the firing rate of neuron $i$ as $\nu_i(\mu) = \bar{\nu}_i + \Delta\nu_i(\mu)$ where $\bar{\nu}_i$ is the mean firing rate of neuron $i$ averaged across all winning patterns and $\langle \Delta\nu_i \rangle_{\mu \in C_i} = 0$. We assume that the absolute size of $\Delta\nu_i$ is small, i.e., $\langle (\Delta\nu_i)^2 \rangle_{\mu \in C_i} \ll (\bar{\nu}_i)^2$. Linearization of *Equation (8)* around $\bar{\nu}_i$ yields:

$$||\vec{w}_i|| = \frac{\gamma}{\beta} \frac{G_1(\bar{\nu}_i)}{G_2(\bar{\nu}_i)} \langle \cos(\alpha) \rangle_{\mu \in C_i} + \frac{\gamma}{\beta} \frac{G_1'(\bar{\nu}_i)}{G_2(\bar{\nu}_i)} \langle \cos(\alpha)\Delta\nu_i \rangle_{\mu \in C_i} \tag{11}$$

Elementary geometric arguments for a neuron model with monotonically increasing frequency–current curve yield that the value of $\langle \cos(\alpha)\Delta\nu_i \rangle_{\mu \in C_i}$ is positive (or zero) because an increase in the angle α lowers both the cosine and the firing rate, giving rise to a positive correlation. Since we are interested in a lower bound, we can therefore drop the term proportional to $G_1'$ and evaluate the ratio $G_1/G_2$ to find:

$$||\vec{w}_i|| \geqslant \frac{\gamma}{\beta} \frac{1}{(\bar{\nu}_i)^2} \langle \cos(\alpha) \rangle_{\mu \in C_i} \geqslant \frac{\gamma}{\beta} \frac{1}{(\nu_{\max})^2} \cos(\hat{\alpha}) \tag{12}$$

where $\nu_{\max}$ is the maximal firing rate of a DGC and $\hat{\alpha} = \max_{\mu \in C_i}\{\alpha\}$ is the angle of the winning pattern that has the largest angle with the weight vector. The first bound is tight and is reached if neuron $i$ is winner for only two patterns.

To summarize we find that the length of the weight vector remains bounded in a narrow range. Hence, for a reasonable distribution of input patterns and weight vectors, the value of $||\vec{w}_i||$ is similar for different neurons $i$, so that the weight vector will have, after convergence, similar lengths for all DGCs that are winners for at least one pattern. In our simulations with the MNIST data set, we find that the length of feedforward weight vectors lies in the range between 9.3 and 11.1 across all responsive neurons with a mean value close to 10; *Figure 2e*.

## Early maturation phase

During the early phase of maturation, the GABAergic input onto a newborn DGC with index $l$ has an excitatory effect. In the model, it is implemented as follows: $w_{lk}^{EI} = -w_{EI} > 0$ with probability $p_{EI}$ for any interneuron $k$ and $w_{lk}^{EI} = 0$ otherwise (no connection). Since newborn cells do not project yet onto inhibitory neurons (*Temprana et al., 2015*), we have $w_{kl}^{IE} = 0 \; \forall l$. Newborn DGCs are known to have enhanced excitability (*Schmidt-Hieber et al., 2004*; *Li et al., 2017*), so their threshold is kept at $b_l = 0 \; \forall l$. Because the newborn model DGCs receive lateral excitation via interneurons and their thresholds are zero during the early phase of maturation, the lateral excitatory GABAergic input is always sufficient to activate them. Hence, if the firing rate of a newborn DGC exceeds the LTP threshold θ, the feedforward weights grow toward the presented input pattern, *Equation (1)*.

Presentation of all patterns of the data set once (one epoch) is sufficient to reach convergence of the feedforward weights onto newborn DGCs. We define the end of the first epoch as the end of the early phase, i.e., simulation of one epoch of the model corresponds to about 3 weeks of biological time.

## Late maturation phase

During the late phase of maturation (starting at about 3 weeks [*Ge et al., 2006*]), the GABAergic input onto newborn DGCs switches from excitatory to inhibitory. In terms of our model, it means that all existing $w_{lk}^{EI}$ connections switch their sign to $w_{EI} < 0$. Furthermore, since newborn DGCs develop lateral connections to inhibitory neurons in the late maturation phase (*Temprana et al., 2015*), we set $w_{kl}^{IE} = w_{IE}$ with probability $p_{IE}$, and $w_{kl}^{IE} = 0$ otherwise. The thresholds of newborn DGCs are updated after presentation of pattern μ at time $n \cdot T$ ($b_l^{(n)} = b_l^{(n-1)} + \Delta b_l$) according to

$\Delta b_l = \eta_b(\nu_l - \nu_0)$, where $\nu_0$ is a reference rate and $\eta_b$ a learning rate, to mimic the decrease of excitability as newborn DGCs mature (**Table 1**, Biologically plausible network). Therefore, the distribution of firing rates of newborn DGCs is shifted to the left (toward lower firing rates) at the end of the late phase of maturation compared to the early phase of maturation (**Figure 2c,d**). A sufficient condition for a newborn DGC to win the competition upon presentation of patterns of the novel cluster is that the scalar product between a pattern of the novel cluster and the feedforward weight vector onto the newborn DGC is larger than the scalar product between the pattern of the novel cluster and the feedforward weight vector onto any of the mature DGCs. Analogous to the early phase of maturation, presentation of all patterns of the data set once (one epoch) is sufficient to reach convergence of the feedforward weights onto newborn DGCs. We therefore consider that the late phase of maturation has been finished after one epoch.

## Input patterns

Two different sets of input patterns are used. Both data sets have a number $K$ of clusters and several thousands of patterns per cluster. As a first data set, we use the MNIST $12 \times 12$ patterns (**Lecun et al., 1998**) ($N_{EC} = 144$), normalized such that the L2-norm of each pattern is equal to 1. Normalization of inputs (be it implemented algorithmically as done here or by explicit inhibitory feedback) ensures that, once weight growth due to synaptic plasticity has ended and weights have stabilized, the overall strength of input onto DGCs is approximately identical for all cells (see Section Direction and length of the weight vector). Equalized lengths of weight vectors are, in turn, an important feature of classic soft or hard competitive networks (**Kohonen, 1989**; **Hertz et al., 1991**). The training set contains approximately 6000 patterns per digit, while the testing set contains about 1000 patterns per digit (**Figure 1d**). Both training patterns and test patterns contain a large variety of different writing styles indicating that the clusters of input patterns for each class are broadly distributed around their center of mass.

As a second data set, we use handmade artificial patterns designed such that the distance between the centers of any two clusters, or in other words their pairwise similarity, is the same. All clusters lie on the positive quadrant of the surface of a hypersphere of dimension $N_{EC} - 1$. The cluster centers are Walsh patterns shifted along the diagonal (**Figure 5b**):

$$\begin{aligned}
\vec{P}^1 &= \frac{1}{c_0}(1+\xi, 1-\xi, 1+\xi, 1-\xi, ..., 1+\xi, 1-\xi, 1+\xi, 1-\xi) \\
\vec{P}^2 &= \frac{1}{c_0}(1+\xi, 1+\xi, 1-\xi, 1-\xi, ..., 1+\xi, 1+\xi, 1-\xi, 1-\xi) \\
&... \\
\vec{P}^K &= \frac{1}{c_0}(1+\xi, 1+\xi, 1+\xi, 1+\xi, ..., 1-\xi, 1-\xi, 1-\xi, 1-\xi)
\end{aligned} \tag{13}$$

with $|\xi| < 1$ a parameter that determines the spacing between clusters. $c_0$ is a normalization factor to ensure that the center of mass of all clusters has an L2-norm of 1:

$$c_0 = \sqrt{N_{EC}(1+\xi^2)}. \tag{14}$$

The number of input neurons $N_{EC}$ is $N_{EC} = 2^K$. The scalar product, and hence the angle $\Omega$, between the center of mass of any pair of clusters $k$ and $l$ ($k \neq l$) is a function of $\xi$ (**Figure 5a**):

$$\vec{P}^k \cdot \vec{P}^l = \frac{1}{1+\xi^2} = \cos(\Omega) \tag{15}$$

We define the pairwise similarity $s$ of two clusters as: $s = 1 - \xi$. Highly similar clusters have a large $s$ due to the small distance between their centers (hence a small $\xi$).

To make the artificial data set comparable to the MNIST $12 \times 12$ data set, we choose $K = 7$, so $N_{EC} = 128$, and we generate 6000 noisy patterns per cluster for the training set and 1000 other noisy patterns per cluster for the testing set. Since our noisy high-dimensional input patterns have to be symmetrically distributed around the centers of mass $\vec{P}^k$, yet lie on the hypersphere, we have to use an appropriate sampling method. The patterns $\vec{x}^{\mu(k)}$ of a given cluster $k$ with center of mass $\vec{P}^k$ are thus sampled from a Von Mises–Fisher distribution (**Mardia and Jupp, 2009**):

$$\vec{x}^{\mu(k)} \sim \left(\sqrt{1-a^2}\right)\vec{\zeta} + a\vec{P}^k \tag{16}$$

with $\vec{\zeta}$ an L2-normalized vector taken in the space orthogonal to $\vec{P}^k$. The vector $\vec{\zeta}$ is obtained by performing the singular-value decomposition of $\vec{P}^k$ ($U\Sigma V^* = \vec{P}^k$) and multiplying the matrix $U$ (after removing its first column), which corresponds to the left-singular vectors in the orthogonal space to $\vec{P}^k$, with a vector whose elements are drawn from the standard normal distribution. Then the L2-norm of the obtained pattern is set to 1, so that it lies on the surface of the hypersphere. A rejection sampling scheme is used to obtain $a$ (**Mardia and Jupp, 2009**). The sample $a$ is kept if $\kappa a + (N_{EC}-1)\ln(1-\psi a) - c \geqslant \ln(u)$, with $\kappa$ a concentration parameter, $\psi = \frac{1-b}{1+b}$, $c = \kappa\psi + (N_{EC}-1)\ln(1-\psi^2)$, $u$ drawn from a uniform distribution $u \sim U[0,1]$, $a = \frac{1-(1+b)z}{1-(1-b)z}$, $b = \frac{N_{EC}-1}{\sqrt{4\kappa^2 + (N_{EC}-1)^2} + 2\kappa}$, and $z$ drawn from a beta distribution $z \sim \mathcal{Be}(\frac{N_{EC}-1}{2}, \frac{N_{EC}-1}{2})$.

The concentration parameter $\kappa$ characterizes the spread of the distribution around the center $\vec{P}^k$. In the limit where $\kappa \to 0$, sampling from the Von Mises–Fisher distribution becomes equivalent to sampling uniformly on the surface of the hypersphere, so the clusters become highly overlapping. In dimension $N_{EC} = 128$, if $\kappa > 10^3$, the probability of overlap between clusters is negligible. We use a value $\kappa = 10^4$.

## Classification performance (readout network)

It has been observed that classification performance based on DGC population activity is a good proxy for behavioral discrimination (**Woods et al., 2020**). Hence, to evaluate whether the newborn DGCs contribute to the function of the dentate gyrus network, we study classification performance. Once the feedforward weights have been adjusted upon presentation of many input patterns from the training set (Section Plasticity rule), we keep them fixed and determine classification on the test set using artificial readout units (RO).

To do so, the readout weights ($w_{ki}^{RO}$ from model DGC $i$ to readout unit $k$) are initialized at random values drawn from a uniform distribution: $w_{ki}^{RO} \sim \sigma\mathcal{U}(0,1)$, with $\sigma = 0.1$. The number of readout units, $N_{RO}$, corresponds to the number of learned classes. To adjust the readout weights, all patterns of the training data set that belong to the learned classes are presented one after the other. For each pattern $\vec{x}^{\mu}$, we let the firing rate of the DGCs converge (values at convergence: $\nu_i^{\mu}$). The activity of a readout unit $k$ is given by:

$$\nu_k^{RO,\mu} = g\left(I_k^{RO,\mu}\right) = g\left(\sum_{i=1}^{N_{DGC}} w_{ki}^{RO}\nu_i^{\mu}\right) \tag{17}$$

As we aim to assess the performance of the network of DGCs, the readout weights are adjusted by an artificial supervised learning rule. The loss function, which corresponds to the difference between the activity of the readout units and a one-hot representation of the corresponding pattern label (**Hertz et al., 1991**),

$$L(W^{RO}) = \frac{1}{2}\sum_{k=1}^{N_{RO}} (L_k^{\mu} - \nu_k^{RO,\mu})^2 \tag{18}$$

with $L_k^{\mu}$ the element $k$ of a one-hot representation of the correct label of pattern $\vec{x}^{\mu}$, is minimized by stochastic gradient descent:

$$\Delta w_{ki}^{RO,\mu} = \eta(L_k^{\mu} - \nu_k^{RO,\mu})g'\left(I_k^{RO,\mu}\right)\nu_i^{\mu}. \tag{19}$$

The readout units have a rectified hyperbolic tangent frequency-current curve: $g(x) = \tanh\left(2[x]_+\right)$, whose derivative is: $g'(x) = 2\left(1 - \left(\tanh\left(2[x]_+\right)\right)^2\right)$. We learn the weights of the readout units over 100 epochs of presentations of all training patterns with $\eta = 0.01$, which is sufficient to reach convergence.

Thereafter, the readout weights are fixed. Each test set pattern belonging to one of the learned classes is presented once, and the firing rates of the DGCs are let to converge. Finally, the activity of

the readout units $\nu_k^{RO,\mu}$ is computed and compared to the correct label $L_k^{\mu}$ of the presented pattern. If the readout unit with the highest activity value is the one that represents the class of the presented input pattern, the pattern is said to be correctly classified. Classification performance is given by the number of correctly classified patterns divided by the total number of test patterns of the learned classes.

## Control cases

In our standard setting, patterns from a third digit are presented to a network that has previously only seen patterns from two digits. The question is whether neurogenesis helps when adding the third digit. We use several control cases to compare with the neurogenesis case. In the first control case, all three digits are learned in parallel (*Figure 3a*, control 1). In the two other control cases, we either keep all feedforward connections toward the DGCs plastic (*Figure 3c*, control 3) or fix the feedforward connections for all selective DGCs but keep unselective neurons plastic (as in the neurogenesis case) (*Figure 3b*, control 2). However, in both instances, the DGCs do not mature in the two-step process induced by the GABA-switch that is part of our model of neurogenesis.

## Pretraining with two digits

As we are interested by neurogenesis at the adult stage, we pretrain the network with patterns from two digits, such that it already stores some memories before neurogenesis takes place. To do so, we randomly initialize the weights from EC neurons to DGCs: they are drawn from a uniform distribution ($w_{ij} \sim U[0, 1]$). The L2-norm of the feedforward weight vector onto each DGC is then normalized to 1, to ensure fair competition between DGCs during learning. Then we present all patterns from digits 3 and 4 in random order, as many times as needed for convergence of the weights. During each pattern presentation the firing rates of the DGCs are computed (Section Network architecture and neuronal dynamics) and their feedforward weights are updated according to our plasticity rule (Section Plasticity rule). We find that we need approximately 40 epochs for convergence of the weights and use 80 epochs to make sure that all weights are stable. At the end of pretraining, our network is considered to correspond to an adult stage, because some DGCs are selective for prototypes of the pretrained digits (*Figure 1e*).

## Projection on pairwise discriminatory axes

To assess how separability of the DGC activation patterns develops during the late phase of maturation of newborn DGCs, we project the population activity onto axes which are optimized for pairwise discrimination (patterns from digit 3 versus patterns from digit 5, 4 versus 5, and 3 versus 4). Those axes are determined using Fisher linear discriminant analysis, as explained below.

We determine the vector of DGC firing rates, $\vec{\nu}$, at the end of the late phase of maturation of newborn DGCs upon presentation of each pattern, $\vec{x}$, from digits 3, 4, and 5 of the training MNIST dataset. The mean activity in response to all training patterns $\mu$ from digit $m$, $\vec{\mu}_m = \frac{1}{N_m}\sum_{\mu \in m} \vec{\nu}^\mu$, is computed for each of the three digits ($N_m$ is the number of training patterns of digit $m$). The pairwise Fisher linear discriminant is defined as the linear function $\vec{w}^T \vec{\nu}$ that maximizes the distance between the means of the projected activity in response to two digits (e.g. $m$ and $n$), while normalizing for within-digit variability. The objective function to maximize is thus given as:

$$J(w) = \frac{w^T S_B w}{w^T S_W w} \qquad (20)$$

with $S_B = (\vec{\mu}_m - \vec{\mu}_n)(\vec{\mu}_m - \vec{\mu}_n)^T$ the between-digit scatter matrix, and $S_W = \Sigma_m + \Sigma_n$ the within-digit scatter matrix ($\Sigma_m$ is the covariance matrix of the DGC activity in response to pattern of digit $m$, and $\Sigma_n$ is the covariance matrix of the DGC activity in response to pattern of digit $n$). It can be shown that the direction of the optimal discriminatory axis between digit $m$ and $n$ is given by the eigenvector of $S_W^{-1} S_B$ with the corresponding largest eigenvalue.

We arbitrarily set 'axis 1' as the optimal discriminatory axis between digit 3 and digit 5, 'axis 2' as the optimal discriminatory axis between digit 4 and digit 5, and 'axis 3' as the optimal discriminatory axis between digit 3 and digit 4. For each of the three discriminatory axes, we define its origin (i.e. projection value of 0) as the location of the average projection of all training patterns of the three

digits on the corresponding axis. *Figure 4* represents the projections of DGC activity upon presentation of testing patterns at the end of the early and late phase of maturation of newborn DGCs onto the above-defined axes.

## Statistics

In the main text, we present a representative example with three digits from the MNIST data set (3, 4, and 5). It is selected from a set of 10 random combinations of three different digits. For each combination, one network is pretrained with two digits for 80 epochs. Then the third digit is added and neurogenesis takes place (one epoch of early phase of maturation, and one epoch of late phase of maturation). Furthermore, another network is pretrained directly with the three digits for 80 epochs. Classification performance is reported for all combinations (*Supplementary file 1*).

## Simplified rate network

We use a toy network and the artificial data set to determine whether our theory of integration of newborn DGCs can explain why adult dentate gyrus neurogenesis helps for the discrimination of similar, but not for distinct patterns.

The rate network described above is simplified as follows. We use $K$ DGCs for $K$ clusters. Their firing rate $\nu_i$ is given by:

$$\tau_m \frac{\mathrm{d}\nu_i}{\mathrm{d}t} = -\nu_i + \mathcal{H}(I_i - b_i) \tag{21}$$

where $\mathcal{H}$ is the Heaviside step function. As before, $b_i$ is the threshold, and $I_i$ the total input toward neuron $i$:

$$I_i = \sum_{j=1}^{N_{EC}} w_{ij} x_j + \sum_{k \neq i}^{N_{DGC}} w_{rec} \nu_k \tag{22}$$

with $x_j$ the input of presynaptic EC neuron $j$, $w_{ij}$ the feedforward weight between EC neuron $j$ and DGC $i$, and $\nu_k$ the firing rate of DGC $k$. Inhibitory neurons are modeled implicitly: each DGC directly connects to all other DGCs via inhibitory recurrent connections of value $w_{rec}<0$. During presentation of pattern $\vec{x}^\mu$, the firing rates of the DGCs evolve according to *Equation (21)*. After convergence, the feedforward weights are updated: $w_{ij}^{(\mu)} = w_{ij}^{(\mu-1)} + \Delta w_{ij}$. The synaptic plasticity rule is the same as before, see *Equation (1)*, but with the parameters reported in *Table 1* (Simple network). They are different from those of the biologically plausible network because we now aim for a single winning neuron for each cluster. Note that for an LTP threshold $\theta<1$ all active DGCs update their feedforward weights because of the Heaviside function for the firing rate (*Equation 21*).

Assuming a single winner $i^*$ for each pattern presentation, the input (*Equation 22*) to the winner is:

$$I_{i^*} = \vec{w}_{i^*} \cdot \vec{x}, \tag{23}$$

while the input to the losers is:

$$I_i = \vec{w}_i \cdot \vec{x} + w_{\mathrm{rec}}. \tag{24}$$

Therefore, two conditions need to be satisfied for a solution with a single winner:

$$\vec{w}_{i^*} \cdot \vec{x} > b_i \tag{25}$$

for the winner to actually be active, and:

$$\vec{w}_i \cdot \vec{x} + w_{\mathrm{rec}} < b_i \tag{26}$$

to prevent non-winners to become active. The value of $b_i$ in the model is lower in the early phase than in the late phase of maturation to mimic enhanced excitability (*Schmidt-Hieber et al., 2004*; *Li et al., 2017*).

## Similar versus distinct patterns with the artificial data set

Using the artificial data set with $|\xi| < 1$ (*Equation 13*), the scalar product between the centers of mass of two different clusters, given by *Equation (15)*, satisfies: $0.5 \leqslant \frac{1}{1+\xi^2} \leqslant 1$. This corresponds to $0° \leqslant \Omega \leqslant \Omega_{\max} = 60°$.

After stimulation with a pattern $\vec{x}$, it takes some time before the firing rates of the DGCs converge. We call two patterns 'similar' if they activate, at least initially, the same output unit, while we consider two patterns as 'distinct' if they do not activate the same output unit, not even initially. We now show that, with a large concentration parameter $\kappa$, patterns of different clusters are similar if $\xi < \sqrt{\frac{||\vec{w}_i||}{b_i} - 1}$ and distinct if $\xi > \sqrt{\frac{||\vec{w}_i||}{b_i} - 1}$.

We first consider a DGC $i$ whose feedforward weight vector has converged toward the center of mass of cluster $k$. If an input pattern $\vec{x}^{\mu(k)}$ from cluster $k$ is presented, it will receive the following initial input:

$$I_i = \vec{w}_i \cdot \vec{x}^{\mu(k)} = ||\vec{w}_i|| \cdot ||\vec{x}^{\mu(k)}|| \cdot \cos(\vartheta_{kk}) = ||\vec{w}_i|| \cdot \cos(\vartheta_{kk}) \tag{27}$$

where $\vartheta_{kk}$ is the angle between the pattern $\vec{x}^{\mu(k)}$ and the center of mass $\vec{P}^k$ of the cluster to which it belongs. The larger the concentration parameter $\kappa$ for the generation of the artificial data set, the smaller the dispersion of the clusters, and thus the larger $\cos(\vartheta_{kk})$. If instead, an input pattern from cluster $l$ is presented, that same DGC will receive a lower initial input:

$$I_i = \vec{w}_i \cdot \vec{x}^{\mu(l)} = ||\vec{w}_i|| \cdot ||\vec{x}^{\mu(l)}|| \cdot \cos(\vartheta_{kl}) \approx \frac{||\vec{w}_i||}{1 + \xi^2} \tag{28}$$

The approximation holds for a small dispersion of the clusters (large concentration parameter $\kappa$). We note that there is no subtraction of the recurrent input yet because output units are initialized with zero firing rate before each pattern presentation. By definition, similar patterns stimulate (initially) the same DGCs. A DGC can be active for two clusters only if its threshold is:

$$b_i < \frac{||\vec{w}_i||}{1 + \xi^2} \tag{29}$$

Therefore, with a high concentration parameter $\kappa$, patterns of different clusters are similar if $\xi < \sqrt{\frac{||\vec{w}_i||}{b_i} - 1}$, while patterns of different clusters are distinct if $\xi > \sqrt{\frac{||\vec{w}_i||}{b_i} - 1}$.

## Parameter choice

The upper bound of the expected L2-norm of the feedforward weight vector toward the DGCs at convergence can be computed, see *Equation (10)*. With the parameters in *Table 1* (Simple network), the value is $||\vec{w}_i|| \leqslant 1.5$. Moreover, the input patterns for each cluster are highly concentrated; hence, their angle with the center of mass of the cluster they belong to is close to 0, so we have $||\vec{w}_i|| \approx 1.5$. Therefore, at convergence, a DGC selective for a given cluster $k$ receives an input $I_{i^*} = \vec{w}_{i^*} \cdot \vec{x}^{\mu(k)} \approx 1.5$ upon presentation of input patterns $\vec{x}^{\mu(k)}$ belonging to cluster $k$. We choose $b_i = 1.2$ to satisfy *Equation (25)*. Given $b_i$ the threshold value $\xi_{\text{thresh}}$ for which two clusters are similar (and above which two clusters are distinct) can be determined by *Equation (29)* : $\xi_{\text{thresh}} = 0.5$. We created a handmade data set with $\xi = 0.2$ for the case of similar clusters (therefore with similarity $s = 0.8$), and a handmade data set with $\xi = 0.8$ for the distinct case (hence with similarity $s = 0.2$).

Let us suppose that the weights of DGC $i$ have converged and made this cell respond to patterns of cluster $i$. If another DGC $k$ of the network is selective for cluster $k$, cell $i$ gets the input $I_i = \vec{w}_i \cdot \vec{x}^{\mu(k)} + w_{\text{rec}} \approx \frac{1.5}{1+\xi^2} + w_{\text{rec}}$ upon presentation of input patterns $\vec{x}^{\mu(k)}$ belonging to cluster $k \neq i$. Hence, to satisfy *Equation (26)*, we need $w_{\text{rec}} < b_i - \max_\xi \left( \frac{1.5}{1+\xi^2} \right) \approx -0.24$. We set $w_{\text{rec}} = -1.2$.

Furthermore, a newborn DGC is born with a null feedforward weight vector so that at birth, its input consists only of the indirect excitatory input from mature DGCs, which vanishes if all DGCs are quiescent and takes a value $I_i = -w_{\text{rec}} > 0$ if a mature DGC responds to the input. For the feedforward weight vector to grow, the newborn cell $i$ needs to be active. This could be achieved through spontaneous activity that could be implemented by setting the intrinsic firing threshold at birth to a value

$b_{birth} < 0$. In this case, a difference between similar and distinct patterns is not expected. Alternatively, activity of newborn cells can be achieved in the absence of spontaneous activity under the condition $-w_{rec} > b_{birth}$. For the simulations with the toy model, we set $b_{birth} = 0.9$, which leads to weight growth in newborn cells for similar, but not distinct patterns.

## Neurogenesis with the artificial data set

To save computation time, we initialize the feedforward weight vectors of two mature DGCs at two training patterns randomly chosen from the first two clusters, normalized such that they have an L2-norm of 1.5. We then present patterns from clusters 1 and 2 and let the feedforward weights evolve according to *Equation (1)* until they reach convergence.

We thereafter fix the feedforward weights onto the two mature cells and introduce a novel cluster of patterns as well as a newborn DGC in the network. The sequence of presentation of patterns from the three clusters (a novel one and two pretrained ones) is random. The newborn DGC is born with a null feedforward weight vector, and its maturation follows the same rules as before (plastic feedforward weights). In the early phase, GABAergic input has an excitatory effect (*Ge et al., 2006*) and the newborn DGC does not inhibit the mature DGCs (*Temprana et al., 2015*). This is modeled by setting $w_{rec}^{NM} = -w_{rec}$ for the connections from mature to newborn DGC, and $w_{rec}^{MN} = 0$ for the connections from newborn to mature DGCs. The threshold of the newborn DGC starts at $b_{birth} = 0.9$ at birth, mimicking enhanced excitability (*Schmidt-Hieber et al., 2004*; *Li et al., 2017*), and increases linearly up to 1.2 (same threshold as that of mature DGCs) over 12,000 pattern presentations, reflecting loss of excitability with maturation. The exact time window is not critical. In the late phase of maturation of the newborn DGC, GABAergic input switches to inhibitory (*Ge et al., 2006*), and the newborn DGC recruits feedback inhibition onto mature DGCs (*Temprana et al., 2015*). It is modeled by switching the sign of the connection from mature to newborn DGC: $w_{rec}^{NM} = w_{rec}$ and establishing connections from newborn to mature DGCs: $w_{rec}^{MN} = w_{rec}$. Each of the 6000 patterns is presented once during the early phase of maturation and once during the late phase of maturation.

The above paradigm is run separately for each of the two handmade data sets: the one where clusters are similar ($s = 0.8$) and the one where clusters are distinct ($s = 0.2$).

## Analytical computation of the L2-norm and angle

We consider the case where two mature DGCs have learned their synaptic connections, such that the first mature DGC with feedforward weight vector $\vec{w}_1$ is selective for cluster 1 with normalized center of mass $\vec{P}^1$, and the second mature DGC with feedforward weight vector $\vec{w}_2$ is selective for cluster 2 with normalized center of mass $\vec{P}^2$. After convergence, we have $\vec{w}_1 = \langle ||\vec{w}_1|| \rangle \vec{P}^1$ and $\vec{w}_2 = \langle ||\vec{w}_2|| \rangle \vec{P}^2$, where $\langle ||\vec{w}_k|| \rangle$ is the expected L2-norm of the feedforward weight vector onto mature DGC $k$ that is selective for pretrained cluster $k$. In addition, the upper bound for the L2-norm of the weight vectors of the mature DGCs can be determined $\langle ||\vec{w}_1|| \rangle = \langle ||\vec{w}_2|| \rangle \leqslant 1.5$. In our case, we obtain $\langle ||\vec{w}_1|| \rangle = \langle ||\vec{w}_2|| \rangle \approx 1.49$ because of the dispersion of the patterns around their center of mass; hence, we will use this value for the numerical computations below.

We represent the feedforward weight vector $\vec{w}_i$ onto a newborn DGC as an arrow of length $\langle ||\vec{w}_1|| \rangle$ (*Figure 6—figure supplement 1*). We compute analytically its L2-norm at the end of the early phase of maturation of the newborn DGC, as well as its angle φ with the center of mass of the novel cluster $\vec{P}^i$, to confirm the results obtained numerically (*Figure 6*, *Figure 6—figure supplement 1*).

In the early phase of maturation, the feedforward weight vector onto the newborn DGC grows. The norm stabilizes at a higher value in the case of similar patterns ($s = 0.8$, *Figure 6—figure supplement 1*) than in the case of distinct patterns ($s = 0.2$, *Figure 6—figure supplement 1*). It is due to the fact that the center of mass of three *similar* clusters lies closer to the surface of the sphere than the center of mass of two *distinct* clusters (see below). In the late phase of maturation, for similar clusters we observe a slight increase of the L2-norm of the feedforward weight vector onto the newborn DGC concomitantly with the decrease of angle with the center of mass of the novel cluster (*Figure 6—figure supplement 1*), because the center of mass of the novel cluster lies closer to the surface of the sphere than the center of mass of the three clusters.

## Similar clusters

The angle between the center of mass of any pair of similar clusters ($s = 0.8$, $\xi = 0.2$) is given by *Equation (15)*:

$$\Omega_S = \arccos\left(\frac{1}{1 + 0.2^2}\right) \tag{30}$$

Half the distance between the projections of the center of mass of any pair of two similar clusters on a concentric sphere with radius $\langle\|\vec{w}_1\|\rangle$ is given by (*Figure 6—figure supplement 1*):

$$z = \langle\|\vec{w}_1\|\rangle \cdot \sin\left(\frac{\Omega_S}{2}\right) \tag{31}$$

The triangle that connects the centers of masses of the three clusters is equilateral, and $y$ separates one of its angle in two equal parts ($\pi/6$ [rad] each). So the length $y$ can be calculated:

$$y = \frac{z}{\cos\left(\frac{\pi}{6}\right)} \tag{32}$$

Using Pythagoras formula, we can thus determine the expected L2-norm $\langle\|\vec{w}_i\|\rangle$ of the feedforward weight vector onto the newborn DGC at the end of the early phase of maturation:

$$\langle\|\vec{w}_i\|\rangle = \sqrt{\langle\|\vec{w}_1\|\rangle^2 - y^2}, \tag{33}$$

and finally its angle with the center of mass of the novel cluster:

$$\phi = \arccos\left(\frac{\langle\|\vec{w}_i\|\rangle}{\langle\|\vec{w}_1\|\rangle}\right) \tag{34}$$

The numerical values are as follows: $\langle\|\vec{w}_i\|\rangle \approx 1.47$ and $\phi \approx 9.21[°]$, which correspond to the values on *Figure 6—figure supplement 1*.

## Distinct clusters

In the case of distinct patterns ($s = 0.2$, $\xi = 0.8$), the angle between the center of mass of any pair of clusters is given by *Equation (15)*:

$$\Omega_D = \arccos\left(\frac{1}{1 + 0.8^2}\right) > \Omega_S \tag{35}$$

We can directly compute the expected L2-norm of the feedforward weight vector onto the newborn DGC at the end of the early phase of maturation (*Figure 6—figure supplement 1*):

$$\langle\|\vec{w}_i\|\rangle = \langle\|\vec{w}_1\|\rangle \cdot \cos\left(\frac{\Omega_D}{2}\right) \tag{36}$$

We can then calculate the length $z$ between the projection of the center of mass of one of the two pretrained clusters on a concentric sphere with radius $\langle\|\vec{w}_1\|\rangle$ and the feedforward weight vector onto the newborn DGC:

$$z = \langle\|\vec{w}_1\|\rangle \cdot \sin\left(\frac{\Omega_D}{2}\right) \tag{37}$$

Analogous to the similar case, we observe that $y$ separates one angle of the equilateral triangle connecting the projections of the center of mass of the clusters on the sphere in two equal parts, consequently:

$$y = \frac{z}{\tan\left(\frac{\pi}{6}\right)} \tag{38}$$

Finally, the angle between the center of mass of the novel cluster and the feedforward weight vector onto the newborn DGC at the end of the early phase of maturation is:

$$\phi = \arccos\left( \frac{\langle ||\vec{w}_i|| \rangle^2 + \langle ||\vec{w}_1|| \rangle^2 - y^2}{2\langle ||\vec{w}_i|| \rangle \langle ||\vec{w}_1|| \rangle} \right) \tag{39}$$

We obtain the following approximate values: $\langle ||\vec{w}_i|| \rangle \approx 1.34$ and $\phi \approx 47.2[^{\circ}]$, which correspond to the values on *Figure 6—figure supplement 1*. The angle φ is smaller in the similar case than in the distinct case, hence the norm is larger in the similar case, as observed in *Figure 6—figure supplement 1*.

### Effective dimensionality and participation ratio

The effective dimensionality of the input is measured as the participation ratio (PR) defined as $PR = (\mathrm{Tr}(C))^2 / \mathrm{Tr}(C^2)$, where $C$ is the covariance matrix of the input patterns, and $\mathrm{Tr}(C)$ denotes the trace of matrix $C$ (*Mazzucato et al., 2016*; *Litwin-Kumar et al., 2017*).

## Acknowledgements

We thank Josef Bischofberger and Laurenz Wiskott for great discussions and useful remarks, as well as Paul Miller and an anonymous reviewer for constructive comments and suggestions. This research was supported by the Swiss National Science Foundation (no. 200020 184615) and by the European Union Horizon 2020 Framework Program under grant agreement no. 785907 (HumanBrain Project, SGA2).

## Additional information

### Funding

| Funder | Grant reference number | Author |
| --- | --- | --- |
| Swiss National Science Foundation | 200020 184615 | Wulfram Gerstner |
| Horizon 2020 Framework Programme | 785907 | Wulfram Gerstner |

The funders had no role in study design, data collection and interpretation, or the decision to submit the work for publication.

### Author contributions

Olivia Gozel, Conceptualization, Software, Formal analysis, Writing - original draft, Writing - review and editing; Wulfram Gerstner, Conceptualization, Funding acquisition, Writing - review and editing

### Author ORCIDs

Olivia Gozel  https://orcid.org/0000-0003-2223-4097

### Decision letter and Author response

Decision letter https://doi.org/10.7554/eLife.66463.sa1
Author response https://doi.org/10.7554/eLife.66463.sa2

## Additional files

### Supplementary files

• Supplementary file 1. Classification performance for random combinations of digits. The classification performance ($P_0$, $P_1$, $P_2$) is defined as the percentage of correctly classified patterns on the test set. The numbers m n + q (first column) indicate that MNIST digits m and n are used for pretraining (second column); m, n and q are used for pretraining (third column); or m and n are used for pretraining, and patterns from digit q added after neurogenesis (fourth column). $P_2 - P_1$ (last column) is used for evaluating the contribution of neurogenesis to classification performance.

• Supplementary file 2. Comparison of networks with different numbers of inhibitory neurons. The number of excitatory neurons is $N_{DGC} = 100$ for all three networks, and there are $N_I$ inhibitory neurons. The case with $N_I = 25$ is the one presented in the main text. All other network parameters are unchanged (including $p^*$). Each network is pretrained once with digits 3 and 4. The percentage of active neurons (firing rate > 1 Hz) for each testing pattern of the corresponding digit is given (mean ± standard deviation), as well as the classification performance over all testing patterns from the trained digits.

• Supplementary file 3. Network with 700 DGCs (expansion factor from EC to dentate gyrus of about 5) compared to the case with $N_{DGC} = 100$ as in the main text. All other network parameters are unchanged. Each network is pretrained with digits 3 and 4. Note that only a subset of neurons responsive to digit 3 (or 4) get active (firing rate > 1 Hz) for a given pattern 3 (or 4). Classification performance is evaluated over all test patterns from the trained digits. Top: after pretraining; bottom: late phase, after adding patterns from digit '5'. Either all unresponsive cells (*Figure 4—figure supplement 1*), or only a fraction of these (*Figure 4—figure supplement 2*), have been replaced by newborn model cells. For the network with 700 DGCs, about 16-18% of DGCs are activated upon presentation of a digit 3 or 4 or 5 (about 112-126 model DGCs). If 119 newborn DGCs are plastic during presentation of the novel digit 5 (middle column), these can become selective for prototypes of digit 5 (*Figure 4—figure supplement 2*) yielding a good classification performance while keeping 156 unresponsive DGCs available for future tasks. If only 35 newborn DGCs are available, classification performance is lower (right column).

• Supplementary file 4. Classification performance with plastic mature DGCs. Top: Using the main neurogenesis network with $N_{DGC} = 100$ DGCs, we keep the learning rate of newborn DGCs at $\eta = 0.01$, but now set the learning rate of mature DGCs to nonzero values ($\eta_{\text{mature}} > 0$) throughout maturation of newborn DGCs. This enables us to vary the level of remaining plasticity in mature DGCs. The number of newborn DGCs that undergo neurogenesis ($N_{\text{newborn}} = 21$) is the same as in the main text. Overall classification performance for digits 3, 4, and 5 ($P$) is computed at the end of the late phase of maturation of newborn DGCs, as well as the classification performance for digit 3 ($P_3$), digit 4 ($P_4$) and digit 5 ($P_5$). Bottom: Same with the extended neurogenesis network with $N_{DGC} = 700$. The number of newborn DGCs is either set to $N_{\text{newborn}} = 119$ (corresponding to 17% of newborn DGCs), or $N_{\text{newborn}} = 35$ (corresponding to 5% of newborn DGCs). The results with $\eta_{\text{mature}} = 0$ from the main text are repeated here for convenience.

• Supplementary file 5. Comparison of the neurogenesis model and the random initialization model for different input dimensionalities. The simplified model with $s = 0.8$ (similar input clusters) is used. Pretraining with two clusters and subsequent learning of a novel cluster 3 (Neuro.) was performed in the same way as reported in the main text. After full maturation of the newborn DGC (two epochs), its weights were fixed, and patterns of a novel cluster 4 were introduced as well as another newborn DGC, and so on until all seven clusters were learned. Reconstruction errors were computed at the end of learning of all seven clusters, and compared with two cases where newborn DGCs do not undergo a two-phase maturation during their 2 epochs of learning, always stay plastic, and are born with a randomly initialized feedforward weight vector: one where the L2-norm of the weight vector starts at a low value of 0.1 (RandInitL.), and one where the L2-norm starts at 1.5, which is the upper bound for the length of the weight vector (RandInitH.). We compare the reconstruction error between the neurogenesis model and the random initialization models for different values of the effective input dimensionality (PR), which depends on the concentration parameter (κ) used when creating the artificial dataset. The results with the dataset used in the main text ($\kappa = 10^4$, PR =11) are reported here for comparison.

• Transparent reporting form

### Data availability

Simulation and plotting scripts can be found at: https://github.com/ogozel/NeurogenesisModel (copy archived at https://archive.softwareheritage.org/swh:1:rev:e46f2dfc10c21d69ac057f31c5800f46644b004a).

The following previously published dataset was used:

| Author(s) | Year | Dataset title | Dataset URL | Database and Identifier |
|---|---|---|---|---|
| LeCun Y, Cortes C, Burges CJC | 1999 | The MNIST database of handwritten digits | http://yann.lecun.com/exdb/mnist/ | THE MNIST DATABASE, yann.lecun.com/exdb/mnist/ |

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
