## [Decision Letter]

**Acceptance summary:**

This paper demonstrates through theoretical modelling how the switch from excitatory to inhibitory signaling occurring in new born neurons can aid the integration of new neurons into the existing neural circuit. The modelling analysis also analyzes how this can aid the temporal integration of relevant memories.

**Decision letter after peer review:**

Thank you for submitting your article "A functional model of adult dentate gyrus neurogenesis" for consideration by *eLife*. Your article has been reviewed by 2 peer reviewers, and the evaluation has been overseen by a Reviewing Editor and John Huguenard as the Senior Editor. The following individuals involved in review of your submission have agreed to reveal their identity: Paul Miller (Reviewer #1).

Essential Revisions:

1. Provide analysis of the model where mature neurons also exhibit plasticity but at reduced levels.

2. Examine how network behaves when inputs have different statistics.

*Reviewer #1:*

The authors propose a role for newborn cells in the dentate gyrus that relies upon their input from interneurons being initially excitatory before switching to inhibitory as the cells mature. The computational modeling and accompanying analyses show how, when receiving only excitatory input, the newborn cells become responsive to stimuli similar to those that already cause high responses in other cells, but then, following the developmental switch such that they receive inhibitory input, those cells then gain responses to novel, but similar, inputs. In a simplified model, the authors are able to quantify the criterion of "sufficient similarity", such that if the novel inputs are not similar enough to the original ones the newborn cells to not gain responses to them. The authors demonstrate that such only when newborn cells are incorporated into the network and respond to the novel stimuli, can those stimuli be categorized by the network, as necessary for them to be recognized.

A major achievement of the paper is to identify a role in information processing for the developmental shift in reversal potential of chloride ions. Such a role is well supported by the results in the paper.

As in all modeling papers, some choices must be made so as to simplify the system to render it tractable and its behavior understandable. Some of the choices could be better justified or discussed, as highlighted below.

The normalization of inputs such that the L2-norm is fixed seems rather unusual, and is not clearly the actual impact of feedforward inhibition. It would be nice to know whether this feature of the input vector is important. Would it matter if the L1-norm were used for normalization, or if normalization were not precise? Perhaps a comment on the importance of this could be made, as well as a justification for the choice.

Throughout the manuscript, the authors employ a homeostatic term in the postsynaptic firing rate. The term is called "heterosynaptic" by the authors, but strictly it is not, since it does not depend on other presynaptic inputs. Rather, the plasticity rule effects "firing rate homeostasis" and is implemented in a manner similarly to Renart, Song, and Wang, (2003). I think this should be mentioned at a minimum and perhaps entirely renamed throughout the manuscript.

The authors consider the ability of the network to discriminate novel patterns as the newborn cells gain responses to the novel patterns. I assume that the formation of new responses arises when the novel patterns are presented randomly amidst the set of previously learned patterns. It would be valuable to see if there are prediction of any differences in behavior if the novel patterns were presented alone, or if there are any impacts of different manners of interspersing learned patterns with novel patterns.

Lines 68-91 contain a lot of details of the circuitry, many of which are not included in the model. It would be helpful to have a figure showing the full circuit based on the information written (which is rather hard to take in through one reading) and beside it to include a figure of the model circuit so the reader can easily see what is being simplified and omitted.

Lines 118-119: I think you should mention here or in the Discussion that you have selected a specific set of synapses to undergo plasticity-that is, if I understand correctly, you have ignored any plasticity with the Dentate Gyrus.

Line 167-8, Equation 1: This equation appears to be different from that of Equation6 in the methods. In particular the "HET" function depends on postsynaptic rate-cubed, not just the difference between rate and threshold as suggested here. Why not just write the exact equation and indicate/describe the behavior of each term?

Line 260-261: The terminology is a bit confusing, as activation is not clearly a "change in membrane potential" but a change in firing rate, so has different units to the reversal potential. Especially as the membrane potential must be venturing above threshold to produce some spiking activity. Perhaps the criterion is equivalent to "the activity is low enough that the mean membrane potential remains below the reversal potential of the chloride channels"?

Line 272-3: The statement about the switch in excitability here assumes we already know it, though it is described in the methods much later. Perhaps this sort of issue is inevitable in journals where methods are placed after results, but it would be better if the order were reversed!

Line 320: I see no justification for a one-way t-test. I think they should be two-way unless it is only possible a priori for a change in one direction.

Line 338: The mention of fixing one set of inputs arises out of nowhere without justification – though that justification comes later as this is just one of two controls. I think it would be better with the order reversed. Or, at least when it is first mentioned here, please be clear why this was chosen, as – I assume – the feedforward weights to selective cells are not fixed in the main set of results. If feedforward weights to selective cells are fixed in the earlier sections, then it should be clearly mentioned, as I did not notice it.

Line 364: Following the previous comment, this line suggests that feedforward weights are not fixed in your primary results with good discrimination. Please clarify if this statement is constrained to the networks without neurogenesis.

*Reviewer #2:*

Gozel and Gerstner investigated the functional role of adult neurogenesis in the dentate gyrus using simulations and mathematical analysis of a computational model. The novelty of the paper compared to numerous previous studies in the field is the inclusion of the GABAergic switch from excitation to inhibition of new neurons during the maturation process. So far this has been overlooked in the computational literature. G&G propose an elegant and potentially interesting idea for how the two phase maturation process could be functionally beneficial for an animal tasked with discriminating stimuli, and would be the first to recapitulate the experimental finding of adult neurogenesis contributing to pattern separation of similar but not distinct stimuli.

However, my assessment is that the current model simulations and analysis are not sufficient to support for the claims made in the paper. Furthermore, the main experimental finding that can be understood based on this modeling work is emergence pattern separation for similar but not distinct stimuli. While interesting, this is rather technical, and may depend quite strongly on the details of the model.

1. The input stimuli from the MNIST dataset presented to the network are low dimensional to a very good approximation ("3", "4", "5"), in contrast to the type of stimuli a real network would be presented with which are expected to be high-dimensional.

1.1. In the model analyzed, the narrowness of the distribution of synaptic weight vector norms is important for network stability. This narrow distribution could at least in part be inherited from the low dimensionality of stimuli (all "3's" have large overlap with the "average 3"). If the overlaps of different stimuli are broadly distributed, so will the distribution of how many input patterns each neuron is a "winner" in. It is important to test this stability in more realistic stimulus ensembles, perhaps by controlling the width of the overlap distribution using the binary model the authors present towards the end of the paper.

1.2. The authors claim that synapses of newborn DGCs starting the maturation process from 0 is important for solving the problem of unresponsive neurons. The reason is that during this phase the synaptic weight vector becomes aligned with a specific direction of the input space. It is possible that unresponsive neurons are stuck in a local minimum (like the case with no neurogenesis) precisely because stimuli (and overlaps) are narrowly distributed around the mean. If stimuli are more broadly distributed (~higher dimension), the basins of attraction are expected to be more numerous and more shallow. Therefore one may expect the problem of the system getting stuck in a local minimum to be far less severe in this case, and for "control 2" networks to learn well.

2. Setting no plasticity (eta = 0) for mature cells is a very strong assumption. Some protocols (e.g., TBS_2_ in Schmidt-Hieber, 2004; and others in Ge 2007) lead to ~2 fold increase in plasticity in young vs. mature neurons. Since mature neurons significantly outnumber young neurons, the effect of plasticity in mature neurons cannot be neglected altogether, especially since the paper's main focus is on the integration of newborn neurons into the circuit. Given the actual degree of synaptic plasticity in mature neurons (according to the papers that the authors themselves cite) I expect the behavior of the authors model to be much closer to "control 3". To support their claims, I think the authors should show that their network compares favorably to control 3 even if DGCs remain plastic throughout (but to a lesser extent). In this scenario I expect the fraction of neurons that are new at any given time to be much more important than the current model, since the mature part of the network is fixed. Therefore this fraction should also be matched to experiments.

3. It is not clear to me how the two phase maturation process of DGCs would be affected in a scenario where at any given point some DGCs are in the excitatory phase of GABA and others are in the inhibitory phase. This would be expected if there is a continuous stream of new neurons. Would the plasticity of the neurons in the inhibitory phase not interfere with aligning the activity to similar stimuli due to plasticity of neurons in the excitatory phase? If there is interference, would the authors then predict that neurogenesis occurs in waves (i.e., some kind of global signal would coordinate transition from phase 1 to 2 across synapses)?

Is there evidence supporting that?

It seems to me that the calculation in the section "Analytical computation of the L2-norm and angle"--at least in principle--be extended to estimate the interference: the competition due to plasticity of neurons in the inhibitory phase increases the angle phi, and thus slows down the alignment of the weights due to plasticity of neurons in the excitatory phase.

107, Review of functional role of DGCs.

Aljadeff et al., 2015,

Shani-Narkiss et al., 2020

suggest a dynamical role for new neurons.

312, It would be interesting if the advantage of adding newborn neurons stimulated with "5" to a network pretrained with "3" and "4" over a network pretrained with "3", "4", and "5" would persist if some amount of plasticity remains in mature neurons (Figure 3d).

614, It would be good to discuss the possibility that neurotransmitter switch (without neurogenesis) has the same functional role as GABA switch in the current model. See e.g., Li et al., (2020) J Neuroscience.

Furthermore, can this model teach us anything about neurogenesis in the olfactory bulb? Is there an E to I switch there too?

725, Miller and Fumarola may not be the right reference to cite here. This specific nonlinearity (rectified tanh) is not standard and is not included in that paper.

778, Definition of quasi orthogonal is not clear. The inhibitory rates can have fluctuations and temporal dynamics of their own even if the network is assumed to be silent when each stimulus is presented. Therefore inputs might be quasi-orthogonal at one time but not at another. If in this is used just to qualitatively understand the network behavior, this somewhat sloppy definition is ok, but I think this caveat should be mentioned to avoid confusion.

---

## [Author Response]

Essential Revisions:1. Provide analysis of the model where mature neurons also exhibit plasticity but at reduced levels.

We have included a new paragraph in the Results section “Robustness of the model” (lines 473-495) and a Supplementary File 4 to address this point.

2. Examine how network behaves when inputs have different statistics.

We have included a new Results section “The cooperative phase of maturation promotes pattern separation for any dimensionality of input data” (lines 601-678) as well as a new Supplementary File 5 to address this point (and a Method section “Effective dimensionality and participation ratio”, lines 1349-1353, to define our dimensionality measure).

Reviewer #1:The authors propose a role for newborn cells in the dentate gyrus that relies upon their input from interneurons being initially excitatory before switching to inhibitory as the cells mature. The computational modeling and accompanying analyses show how, when receiving only excitatory input, the newborn cells become responsive to stimuli similar to those that already cause high responses in other cells, but then, following the developmental switch such that they receive inhibitory input, those cells then gain responses to novel, but similar, inputs. In a simplified model, the authors are able to quantify the criterion of "sufficient similarity", such that if the novel inputs are not similar enough to the original ones the newborn cells to not gain responses to them. The authors demonstrate that such only when newborn cells are incorporated into the network and respond to the novel stimuli, can those stimuli be categorized by the network, as necessary for them to be recognized.A major achievement of the paper is to identify a role in information processing for the developmental shift in reversal potential of chloride ions. Such a role is well supported by the results in the paper.As in all modeling papers, some choices must be made so as to simplify the system to render it tractable and its behavior understandable. Some of the choices could be better justified or discussed, as highlighted below.The normalization of inputs such that the L2-norm is fixed seems rather unusual, and is not clearly the actual impact of feedforward inhibition. It would be nice to know whether this feature of the input vector is important. Would it matter if the L1-norm were used for normalization, or if normalization were not precise? Perhaps a comment on the importance of this could be made, as well as a justification for the choice.

We thank the reviewer for raising an important point. In our model, it is merely a practical simplification to consider input patterns which all have an L2-norm of 1. Indeed, it ensures that the upper bound of the L2-norm of the feedforward weight vectors onto newborn DGCs is fixed and identical for all newborn DGCs (see Methods, section “Direction and length of the weight vector”).

In competitive networks, it is in general important that the length of the feedforward weight vectors onto different DGCs is similar, otherwise the cells with longer weight vectors would have an unfair advantage and thus have higher probability to win the competition. Since the input is normalized and the weight vector converges to the center of inputs for which it becomes active, it follows that all the final weights vectors are expected to be of similar length. Indeed, in our model, at the end of maturation, newborn DGCs do not get identical weight vector lengths, though the distribution is rather narrow. For unprecise L2-normalization of the inputs, we expect that if the imprecision is uniform over the input space, then our neurogenesis model should still do fine, because the weight vector length of each newborn DGC would depend on input pattern lengths that have the same statistics. However, if the input space has some regions of input patterns that have much higher L2-norm than other regions of input patterns, then we expect the clusters with higher norm to be well represented by model newborn DGCs, while clusters with low L2-norms would be poorly represented, therefore discrimination of input patterns would decrease.

If the L1-norm was used for input normalization, it would yield weight vectors with slightly variable L2-norms non-uniformly distributed in the input space. Without loss of generality, we can consider 2-dimensional inputs for a visual explanation. If the L1-norm of input patterns is fixed to a value R, then inputs whose direction is horizontal (arrowhead at (R,0)) or vertical (arrowhead at (0,R)) will have larger L2-norm (R) than inputs whose direction is diagonal to the xy-plane (arrowhead at (R/2,R/2), so L2-norm=sqrt(½)*R). Therefore, difference between L1 and L2 normalization will show up when one compares diagonally oriented input vectors with inputs aligned with one of the axes.

We added an explanation for the reason for our L2-normalization implementation choice in lines 1033-1039. It reads:

“Normalization of inputs (be it implemented algorithmically as done here or by explicit inhibitory feedback) ensures that, once weight growth due to synaptic plasticity has ended and weights have stabilized, the overall strength of input onto DGCs is approximately identical for all cells (see Section Direction and length of the weight vector). Equalized lengths of weight vectors are, in turn, an important feature of classic soft or hard competitive networks (Kohonen, 1989; Hertz et al., 1991).”

Throughout the manuscript, the authors employ a homeostatic term in the postsynaptic firing rate. The term is called "heterosynaptic" by the authors, but strictly it is not, since it does not depend on other presynaptic inputs. Rather, the plasticity rule effects "firing rate homeostasis" and is implemented in a manner similarly to Renart, Song, and Wang (2003). I think this should be mentioned at a minimum and perhaps entirely renamed throughout the manuscript.

In our manuscript, we used the definition of “heterosynaptic” that can be found in Chistiakova et al., (2014) and Zenke and Gerstner, (2017). It is a homeostatic mechanism but differs from standard experimentally observed synaptic scaling mechanisms (Turrigiano et al., 1998) mainly by the fact that it occurs on a much shorter timescale: “Homeostatic plasticity differs from the heterosynaptic plasticity […] in two important aspects. First, homeostatic plasticity requires nonspecific dramatic changes of neuronal activity over prolonged periods, which are unlikely to happen during everyday life and learning. Second, it operates on a very long time scale, hours and days, and thus cannot counteract runaway dynamics induced within seconds and minutes by Hebbian-type learning rules.” (Chistiakova et al., 2014). In Renart et al., (2003), they model a “homeostatic” mechanism: it has a long characteristic timescale. However, their actual implementation differs: “since what matters for the steady state of the very slow scaling process is the integrated activity of each cell across different stimuli, we have replaced this temporal average by a spatial average carried out over several network simulations run in parallel” (Renart et al., 2003).

In this terminology, “homeostatic” and “heterosynaptic” mechanisms both depend on postsynaptic activity but are independent of presynaptic activity (the “heterosynaptic” term in our synaptic plasticity rule does not depend on the presynaptic firing rate x_j_). Since they do not depend on the identity of the presynaptic neuron, heterosynaptic plasticity affects several synapses in parallel, hence the terminology. For example, a strong presynaptic input at synapse j may cause the postsynaptic neuron to fire at a very high rate. The heterosynaptic term then lowers the strength of other synapses k nonequal j, independent of the presynaptic activity. On the other hand, “homosynaptic” terms are Hebbian: they depend on pre- and postsynaptic activity.

As a personal aside, I (Wulfram) would add that we switched to the term heterosynaptic in my lab after the work of Friedemann Zenke. Whenever I gave a talk and mentioned homeostatic plasticity, all participants immediately had in mind the beautiful work of Turrigiano – but this work considers changes on the time scale of 24 hours. However, Zenke showed that you cannot stabilize firing rates in a recurrent network if the homeostatic mechanism works on the time scale of several hours. We had a meeting at the Royal Academy of Sciences in London some years ago where homeostatic mechanisms were debated controversially. I would like to pull out of the controversy by talking about heterosynaptic plasticity that contributes to firing rate homeostatis. Importantly, the heterosynaptic plasticity that we use is in the same equation (and on the same time scale) as the Hebbian plasticity. Two review papers of Friedemann Zenke give the main arguments for this terminology.

To make this issue clearer to the reader and avoid potential confusion of terminology, we now say in the text always heterosynaptic plasticity inducing rapid homeostatic weight stabilization, for example:

– lines 185-192: “The third term on the right-hand-side of equation (1) implements heterosynaptic plasticity (Chistiakova et al., 2014; Zenke and Gerstner, 2017): whenever the postsynaptic neuron fires at a rate above theta, all weights are downregulated independent of presynaptic activity. It ensures that the weights cannot grow without bounds (Methods). In this sense, the third term has a 'homeostatic' function (Zenke and Gerstner, 2017), yet acts on a time scale faster than experimentally observed homeostatic synaptic plasticity (Turrigiano et al., 1998).”

– lines 240-244: “A detailed mathematical analysis (Methods) shows that heterosynaptic plasticity in equation (1) ensures that the total strength of the receptive field of each selective DGC converges to a stable value which is similar for selective DGCs confirming the homeostatic function of heterosynaptic plasticity (Zenke and Gerstner, 2017).”

– lines 921-925: “Moreover, all weights onto neuron i are downregulated heterosynaptically by an amount that increases supra-linearly with the postsynaptic rate nu_i_, implicitly controlling the length of the weight vector (see below) similar to synaptic homeostasis (Turrigiano et al., 1998) but on a rapid time scale (Zenke and Gerstner, 2017).”

The authors consider the ability of the network to discriminate novel patterns as the newborn cells gain responses to the novel patterns. I assume that the formation of new responses arises when the novel patterns are presented randomly amidst the set of previously learned patterns. It would be valuable to see if there are prediction of any differences in behavior if the novel patterns were presented alone, or if there are any impacts of different manners of interspersing learned patterns with novel patterns.

It is correct that in our implementation, novel patterns are presented randomly amidst the set of previously learned patterns. But an aspect that is even more important (as shown with our simplified network model) is whether novel patterns are, or are not, similar enough to familiar patterns. Let us discuss four situations:

First, if similar novel patterns are presented alone during the early phase of maturation, we expect that the feedforward weight vector onto newborn DGCs will grow directly in the direction of the center of mass of the novel patterns. Indeed, if novel patterns are similar to familiar patterns, they will activate mature DGCs which will indirectly activate the newborn DGCs. Thus, whether novel patterns similar to familiar patterns are presented alone or amidst familiar patterns ultimately makes no difference at the end of maturation.

Second, in the late phase of maturation we expect the feedforward weight vector onto newborn DGCs to represent different prototypes of the novel similar patterns, no matter if only novel patterns are presented or if both novel and familiar patterns are presented. (However, if only familiar patterns are presented during the late competitive phase of maturation, newborn DGCs will never win the competition (because they were selective for novel patterns at the end of the early phase), hence stay silent and not update their weights.)

Third, we show that the timing of introduction of the novel patterns is important (lines 455-472). Introduction of similar novel patterns must be early enough, otherwise feedforward weight vectors onto newborn DGCs grow in a direction which is too far from novel patterns. However, if novel patterns are introduced towards the end of the early phase but are not interspersed with familiar patterns, we expect that a shorter period is sufficient for changing the direction of the weight vector to a direction which is close enough from novel patterns to be able to become selective for them in the late phase of maturation.

Fourth, we expect the L2-norm of the feedforward weight vector onto newborn DGCs to grow faster in the early phase if only familiar patterns are presented initially.

In summary, a blocked presentation of novel patterns that are similar to familiar patterns timed towards the end of the early phase is actually helpful – and an interspersed presentation is actually a scenario which yields slower learning. The advantage of the interspersed presentation is that the timing does not need to be optimized, which is the reason for our implementation choice.

We now specify more clearly our implementation and its rationale (lines 267-271):

“To mimic exposure of an animal to a novel set of stimuli, we now add input patterns from digit 5 to the set of presented stimuli, which was previously limited to patterns of digits 3 and 4. The novel patterns from digit 5 are randomly interspersed into the sequence of patterns from digits 3 and 4; in other words, the presentation sequence was not optimized with a specific goal in mind.”

Lines 68-91 contain a lot of details of the circuitry, many of which are not included in the model. It would be helpful to have a figure showing the full circuit based on the information written (which is rather hard to take in through one reading) and beside it to include a figure of the model circuit so the reader can easily see what is being simplified and omitted.

Thanks for the suggestion. We added a new Figure 1 —figure supplement 1 and refer to it in the main text where we write about the circuitry (lines 71, 92, 141, 216, 854, 860).

Lines 118-119: I think you should mention here or in the Discussion that you have selected a specific set of synapses to undergo plasticity-that is, if I understand correctly, you have ignored any plasticity with the Dentate Gyrus.

Thank you for pointing out the need for more clarity. In our model, the only synapses that are plastic and follow our plasticity rule are those between EC and DGCs. There is no plasticity rule involved within the dentate gyrus: connections are absent or present (with fixed values). However, the connections between newborn DGCs (E) and inhibitory neurons (I) within dentate gyrus still changes as a function of maturation: from no E-to-I and fixed positive I-to-E connections in the early phase to fixed positive E-to-I and fixed negative I-to-E connections in the late phase.

To clarify this important point, we now specify that it is the synaptic connections between EC and newborn DGCs which are plastic, and write in lines 120-123 (Introduction):

“[…] our model uses an unsupervised biologically plausible Hebbian learning rule that makes synaptic connections between EC and newborn DGCs either disappear or grow from small values at birth to values that eventually enable feedforward input from EC to drive DGCs.”

Furthermore, we explicitly mention the lack of plasticity at other synapses within the dentate gyrus in lines 200-202 (Results):

“For simplicity, no plasticity rule was implemented within the dentate gyrus: connections between newborn DGCs and inhibitory cells are either absent or present with a fixed value (see below).”

Line 167-8, Equation 1: This equation appears to be different from that of Equation6 in the methods. In particular the "HET" function depends on postsynaptic rate-cubed, not just the difference between rate and threshold as suggested here. Why not just write the exact equation and indicate/describe the behavior of each term?

We agree that our “simplified” expression ended up being more confusing than self-explanatory. As suggested, we wrote the exact equation in the Results (line 174-175).

Line 260-261: The terminology is a bit confusing, as activation is not clearly a "change in membrane potential" but a change in firing rate, so has different units to the reversal potential. Especially as the membrane potential must be venturing above threshold to produce some spiking activity. Perhaps the criterion is equivalent to "the activity is low enough that the mean membrane potential remains below the reversal potential of the chloride channels"?

Thanks for pointing out the inaccuracy of our terminology, and for the reformulation suggestion. We modified lines 281-285 as follows:

“We assume that in natural settings, the activation of GABA_A_ receptors is low enough that the mean membrane potential remains below the chloride reversal potential at which shunting inhibition would be induced (Heigele et al., 2016). In this regime, the net effect of synaptic activity is hence excitatory.”

Line 272-3: The statement about the switch in excitability here assumes we already know it, though it is described in the methods much later. Perhaps this sort of issue is inevitable in journals where methods are placed after results, but it would be better if the order were reversed!

Even though we mentioned the change in excitability in the Introduction (lines 38-39), we indeed did not say that we were including it in our model. We hope that we solved this issue by now mentioning that we do model the change in excitability of newborn DGCs by modifying their firing threshold early in the Results section, lines 143-148:

“Firing rates are modeled by neuronal frequency-current curves that vanish for weak input and increase if the total input into a neuron is larger than a firing threshold. Since newborn DGCs exhibit enhanced excitability early in maturation (Schmidt-Hieber et al., 2004; Li et al., 2017), the firing threshold of model neurons increases during maturation from a lower to a higher value (Methods).”

Line 320: I see no justification for a one-way t-test. I think they should be two-way unless it is only possible a priori for a change in one direction.

Thanks for your careful reading. It is true that the change in classification performance could be in both directions. However, we were only interested by a potential difference from a zero change: we did not compare the two distributions themselves (a two-way t-test would test if the mean of one distribution is different from the mean of the other distribution). We clarify this in the new version (lines 353-360):

“Across ten simulation experiments, classification performance is significantly higher when a novel ensemble of patterns is learned sequentially by newborn DGCs (P_2_; Supplementary File 1), than if all patterns are learned simultaneously (P_1_; Supplementary File 1). Indeed, the distribution of P_2_-P_1_ for the ten simulation experiments has a mean which is significantly different from zero (Wilcoxon signed rank test: p-val = 0.0020, Wilcoxon signed rank = 55; one-way t-test: p-val = 0.0269, t-stat = 2.6401, df = 9; Supplementary File 1).”

Line 338: The mention of fixing one set of inputs arises out of nowhere without justification – though that justification comes later as this is just one of two controls. I think it would be better with the order reversed. Or, at least when it is first mentioned here, please be clear why this was chosen, as – I assume – the feedforward weights to selective cells are not fixed in the main set of results. If feedforward weights to selective cells are fixed in the earlier sections, then it should be clearly mentioned, as I did not notice it.

Thanks for your comment. In addition to the next comment, it made us realize that our text from section “Mature neurons represent prototypical input patterns” to section “The GABA-switch guides learning of novel representations” was not completely clear and the flow was sometimes maybe misleading. Therefore, we slightly reorganized the text, and added some details on the implementation. Here are the main changes we made:

– We added a sentence to mention that pretraining starts from random (nonzero) weights, and that all DGCs have the same learning rate during pretraining, lines 206-210: “Hence we start with a network that already has strong random EC-to-DGC connection weights (Methods). We then pretrain our network of 100 DGCs using the same learning rule (equation (1), with identical learning rate eta for all DGCs) that we will use later for the integration of newborn cells.”

– We moved the information about setting the learning rate of selective cells to zero (which was previously at the beginning of section “Newborn neurons become selective for novel patterns during maturation”) to the end of section “Mature neurons represent prototypical input patterns” and explicitly state that mature cells are not plastic in our main implementation (but we included results for when they are), now lines 250-256: “After convergence of synaptic weights during pretraining, selective DGCs are considered mature cells. Mature cells are less plastic than newborn cells (Schmidt-Hieber et al., 2004; Ge et al., 2007). So in the following, unless specified otherwise, we set eta=0 in equation (1) for mature cells (feedforward connection weights from EC to mature cells remain therefore fixed). A scenario where mature cells retain synaptic plasticity is also investigated (see Robustness of the model and Supplementary File 4).”

– We state that newborn DGCs in the main neurogenesis model start with zero feedforward weights, lines 261-263: “In our main neurogenesis model, we replace unresponsive model units by plastic newborn DGCs (eta > 0 in equation (1)) which receive lateral GABAergic input but do not receive feedforward input yet (all weights from EC are set to zero).”

– In lines 348-350, we allude that “pretraining with three digits” (our control 1) is implemented in the exact same way as “pretraining with 2 digits”, except that patterns from three digits (instead of two) are presented: “We compare this performance with that of a network where all three digit ensembles are directly simultaneously pretrained starting from random weights (Figure 3a, control1).”

– We moved the paragraph about introducing two novel digits (previously in lines 322-331) to the end of the previous section “Newborn neurons become selective for novel patterns during maturation” (now lines 329-338). We believe that the flow is better in this way, as this paragraph talks about a variant of the main neurogenesis results (where two novel digits instead of one are introduced during maturation of the newborn cells), and not about the pretraining control (as it may have previously suggested).

We slightly modified section “The GABA-switch guides learning of novel representations” (now lines 361-394) to more clearly contrast controls 2 and 3 with the main neurogenesis model in terms of learning rate of the selective cells and the connectivity of the unresponsive cells.

Line 364: Following the previous comment, this line suggests that feedforward weights are not fixed in your primary results with good discrimination. Please clarify if this statement is constrained to the networks without neurogenesis.

We hope that with the reorganization of the text as explained in the previous point, this statement (now at lines 392-393) is now clear. Specifically, this statement applies to control 3: all DGCs keep plastic feedforward weights. However, unresponsive units at the end of pretraining are not replaced by newborn DGCs which undergo maturation. Upon presentation of novel patterns, those unresponsive units have low probability to become selective, because their feedforward weight vector has a low norm, and they probably point outside of the space of presented inputs (otherwise they would have become selective earlier). Therefore, novel patterns are learned by cells that were already selective for (similar) familiar patterns. On the other hand, in the main neurogenesis model, newborn DGCs (previously unresponsive units) learn the novel patterns, while the selectivity of mature DGCs is not overwritten since they are not plastic.

Reviewer #2:Gozel and Gerstner investigated the functional role of adult neurogenesis in the dentate gyrus using simulations and mathematical analysis of a computational model. The novelty of the paper compared to numerous previous studies in the field is the inclusion of the GABAergic switch from excitation to inhibition of new neurons during the maturation process. So far this has been overlooked in the computational literature. G&G propose an elegant and potentially interesting idea for how the two phase maturation process could be functionally beneficial for an animal tasked with discriminating stimuli, and would be the first to recapitulate the experimental finding of adult neurogenesis contributing to pattern separation of similar but not distinct stimuli.However, my assessment is that the current model simulations and analysis are not sufficient to support for the claims made in the paper. Furthermore, the main experimental finding that can be understood based on this modeling work is emergence pattern separation for similar but not distinct stimuli. While interesting, this is rather technical, and may depend quite strongly on the details of the model.1. The input stimuli from the MNIST dataset presented to the network are low dimensional to a very good approximation ("3", "4", "5"), in contrast to the type of stimuli a real network would be presented with which are expected to be high-dimensional.1.1. In the model analyzed, the narrowness of the distribution of synaptic weight vector norms is important for network stability. This narrow distribution could at least in part be inherited from the low dimensionality of stimuli (all "3's" have large overlap with the "average 3"). If the overlaps of different stimuli are broadly distributed, so will the distribution of how many input patterns each neuron is a "winner" in. It is important to test this stability in more realistic stimulus ensembles, perhaps by controlling the width of the overlap distribution using the binary model the authors present towards the end of the paper.1.2. The authors claim that synapses of newborn DGCs starting the maturation process from 0 is important for solving the problem of unresponsive neurons. The reason is that during this phase the synaptic weight vector becomes aligned with a specific direction of the input space. It is possible that unresponsive neurons are stuck in a local minimum (like the case with no neurogenesis) precisely because stimuli (and overlaps) are narrowly distributed around the mean. If stimuli are more broadly distributed (~higher dimension), the basins of attraction are expected to be more numerous and more shallow. Therefore one may expect the problem of the system getting stuck in a local minimum to be far less severe in this case, and for "control 2" networks to learn well.

We thank the reviewer for this important suggestion. Our datasets are indeed rather low-dimensional. To investigate how our network behaves when the input space is higher dimensional, as suggested, we used our simple network and we created new handmade datasets with higher dimensionality. Our dimensionality measure is now defined in a new Method section “Effective dimensionality and participation ratio” (lines 1349-1353). We included those results in Supplementary File 5 and in a new Results section “The cooperative phase of maturation promotes pattern separation for any dimensionality of input data” (lines 601-678):

“Despite the fact that input patterns in our model represent the activity of 144 or 128 model EC cells, the effective dimensionality of the input data was significantly below 100 because the clusters for different input classes were rather concentrated around their respective center of mass. We define the effective input dimensionality as the participation ratio (Mazzucato et al., 2016; Litwin-Kumar et al., 2017) (Methods). Using this definition, the input data of both the MNIST 12x12 patterns from digits 3, 4 and 5 and the seven clusters of the handmade dataset for similar patterns (s=0.8) are relatively low-dimensional (PR=19 out of a maximum of 144, and PR=11 out of a maximum of 128, respectively). We emphasize that in both cases the spread of the input data around the cluster center implies that the effective dimensionality is larger than the number of clusters. In natural settings, we expect the input data to have even higher dimension. Therefore, here we investigate the effect of dimensionality of the input data on our neurogenesis model by increasing the spread around the cluster centers.

We use our simplified network model and create similar artificial datasets (s=0.8) with different values for the concentration parameter kappa (Methods). The smaller the kappa, the broader the distributions around their center of mass, hence the larger the overlap of patterns generated from different cluster distributions. Therefore, we can increase the effective dimensionality of the input by decreasing the concentration parameter kappa. First, as expected from our analytical analysis (Methods), we find that the broader the cluster distributions the smaller the length of the feedforward weight vector onto newborn DGCs (from just below 1.5 with kappa = 10^4^ to about 1.35 with kappa = 6 * 10^2^). Second, we examine the ability of the simplified network to discriminate input patterns coming from input spaces with different dimensionalities. To do so, we compare our neurogenesis model (Neuro.) with a random initialization model (RandInitL.). In both cases, two DGCs are pretrained with patterns from two clusters, as above. Then we fix the weights of the two mature DGCs and introduce patterns from a third cluster as well as a newborn DGC. For the neurogenesis case, after maturation of the newborn DGC we fix its weights (while for the random initialization model we keep them plastic) upon introduction of patterns from a fourth cluster as well as another newborn DGC, and so on until the network contains seven DGCs and patterns from the full dataset of seven clusters have been presented. We compare our neurogenesis model, where each newborn DGC starts with zero weights and undergo a two-phase maturation (1 epoch per phase), with a random initialization model where each newborn DGC is directly fully integrated into the circuit and whose feedforward weight vector is randomly initialized with a length of 0.1 (RandInitL.) and is then learned for 2 epochs.

Since clusters can be highly overlapping, we assess discrimination performance by computing the reconstruction error at the end of training. Reconstruction error is evaluated analogously to classification error, except that the readout layer has the task of an autoencoder: it contains as many readout units as there are input units. Reconstruction error is the mean squared distance between the input vector and the reconstructed output vector based on testing patterns. We observe that for any dimensionality of the input space, even as high as 97-dimensional, the neurogenesis model performs better (has a lower total reconstruction error) than the random initialization model (Supplementary File 5). Indeed, in the neurogenesis case newborn DGCs grow their feedforward weights (from zero) in the direction of presented input patterns in their early cooperative phase of maturation and can later become selective for novel patterns during the competitive phase. In contrast, since the random initialization model has no early cooperative phase, the newborn DGC weight vector does not grow unless an input pattern is by chance well aligned with its randomly initialized weight vector (which is unlikely in a high dimensional input space). We get similar results for a larger initialization of the synaptic weights (e.g., the length of the weight vector at birth is set to 1, results not shown). Importantly, in high input dimensions, the advantage of a larger weight vector length at birth in the random initialization model is overridden by the capability of newborn DGCs to grow their weight vector in the appropriate direction during their early cooperative phase of maturation. Finally, we note that even if the length of the feedforward weight vector onto newborn DGCs is set to 1.5 (RandInitH., Supplementary File 5), which is the upper bound according to our analytical results (Methods), the random initialization model performs worse than the neurogenesis model for low up to relatively high-dimensional input spaces (PR=83, Supplementary File 5) despite its advantage in the competition conferred by the longer weight vector. It is only when input clusters are extremely broad and overlapping that the random initialization model performs similarly to the neurogenesis model (PR=90,97, Supplementary File 5). In other words, a random initialization at full length of weight vectors works well if input data is homogeneously distributed on the positive quadrant of the unit sphere but fails if the input data is clustered in a few directions. Moreover, random initialization requires that synaptic weights are large from the start which is biologically not plausible. In summary, the two-phase neurogenesis model is advantageous because the feedforward weights onto newborn cells can start at arbitrarily small values; their growth is, during the cooperative phase, guided to occur in a direction that is relevant for the task at hand; the final competitive phase eventually enables specialization onto novel inputs.”

Reply to point 1.1: The narrowness of the distribution of synaptic weight vector norms is indeed important. If the norm of a feedforward weight vector onto a particular DGC would be much larger than the norm of all other feedforward weight vectors onto other DGCs, then that particular DGC would be a winner of the competition upon presentation of more heterogeneous input patterns. In other words, that particular neuron would have much broader tuning, while other DGCs would have narrower tuning. However, our model ensures that the L2-norm of the feedforward weight vectors onto newborn DGCs at the end of their maturation reaches a value between a lower bound and an upper bound (see Methods, “Direction and length of the weight vector”). It implies that if input patterns are highly concentrated around a center of mass (for example in our standard handmade artificial dataset where the concentration parameter kappa=10^4^) the L2-norm will end up being higher than if patterns are broadly distributed around their center of mass (for example cases where kappa < 10^4^). The only way for L2-norms to being widely different is if input patterns are heterogeneously distributed: for example if there is a high concentration around a center of mass (CM1) and a low concentration around another (CM2) then the DGC selective for CM1 will have a larger L2-norm than the DGC selective for CM2. We do not expect this to be a problem if the patterns from cluster 2 are far enough from the patterns of cluster 1 in input space. Furthermore, we note that if patterns from cluster 2 are distributed so broadly that they overlap with patterns from concentrated cluster 1, then: (1) those patterns will be classified by the network as belonging to cluster 1, even though they were initially generated from a “broad cluster 2” distribution; and (2) since those patterns now belong to cluster 1, they do not activate the DGC selective for cluster 2 anymore, hence the distribution of patterns that activate DGC2 becomes narrower and the L2-norm of DGC2 increases (and thus gets closer to the one of DGC1).

Reply to point 1.2: Our new results (see section “The cooperative phase of maturation promotes pattern separation for any dimensionality of input data”) shed light on this point. Briefly, our neurogenesis model still performs better than a random initialization model (aka “control 2”) even for relatively high input space dimensionalities, because the early cooperative phase acts as a smart initialization for the growth of the feedforward weight vector onto the newborn DGC in the appropriate direction. The higher the input space dimensionality, the more advantageous this smart initialization, as there is low probability that a randomly initialized feedforward weight vector onto a newborn cell is sufficiently well aligned with input patterns to become selective for them. It is only when classes are extremely broad and overlapping and that the feedforward weight vector starts with a large norm that the random initialization model performs as well as our neurogenesis model. These results agree with the experimental observation that adult dentate gyrus neurogenesis helps for the discrimination of similar patterns, but not distinct patterns. Similar patterns are close to each other's in input space, therefore smaller deeper basins of attraction are needed to discriminate them. On the other hand, distinct patterns are far from each other's in input space, hence larger and shallower basins of attraction are sufficient to discriminate between them.

2. Setting no plasticity (eta = 0) for mature cells is a very strong assumption. Some protocols (e.g., TBS_2_ in Schmidt-Hieber 2004; and others in Ge 2007) lead to ~2 fold increase in plasticity in young vs. mature neurons. Since mature neurons significantly outnumber young neurons, the effect of plasticity in mature neurons cannot be neglected altogether, especially since the paper's main focus is on the integration of newborn neurons into the circuit. Given the actual degree of synaptic plasticity in mature neurons (according to the papers that the authors themselves cite) I expect the behavior of the authors model to be much closer to "control 3". To support their claims, I think the authors should show that their network compares favorably to control 3 even if DGCs remain plastic throughout (but to a lesser extent). In this scenario I expect the fraction of neurons that are new at any given time to be much more important than the current model, since the mature part of the network is fixed. Therefore this fraction should also be matched to experiments.

The reviewer raises an interesting point. We addressed this concern with new simulations with the main and extended networks. We added a new Supplementary File 4, and report the new results in section “Robustness of the model” (lines 473-495):

“Finally, in our neurogenesis model, we have set the learning rate of mature DGCs to zero despite the observation that mature DGCs retain some plasticity (Schmidt-Hieber et al., 2004; Ge et al., 2007). We therefore studied a variant of the model in which mature DGCs also exhibit plasticity. First, we used our main model with 100 DGCs and 21 newborn DGCs. The implementation was identical, except that the learning rate of the mature DGCs was kept at a nonzero value during the maturation of the 21 newborn DGCs. We do not observe a large change in classification performance, even if the learning rate of the mature cells is the same as that of newborn cells (Supplementary File 4). Second, we used our extended network with 700 DGCs to be able to investigate the effect of plastic mature DGCs while having a proportion of newborn cells matching experiments. We find that with 35 newborn DGCs (corresponding to the experimentally reported fraction of about 5%), plastic mature DGCs (with a learning rate half of that of newborn cells) improve classification performance (Supplementary File 4). This is due to the fact that several of the mature DGCs (that were previously selective for '3's or '4's) become selective for prototypes of the novel digit 5. Consequently, more than the 35 newborn DGCs specialize for digit 5, so that digit 5 is eventually represented better by the network with mature cell plasticity than the standard network where plasticity is limited to newborn cells. Note that those mature DGCs that had earlier specialized on writing styles of digits 3 or 4 similar to a digit 5 are most likely to retune their selectivity. If the novel inputs were very distinct from the pretrained familiar inputs, mature DGCs would be unlikely to develop selectivity for the novel inputs.”

3. It is not clear to me how the two phase maturation process of DGCs would be affected in a scenario where at any given point some DGCs are in the excitatory phase of GABA and others are in the inhibitory phase. This would be expected if there is a continuous stream of new neurons. Would the plasticity of the neurons in the inhibitory phase not interfere with aligning the activity to similar stimuli due to plasticity of neurons in the excitatory phase? If there is interference, would the authors then predict that neurogenesis occurs in waves (i.e., some kind of global signal would coordinate transition from phase 1 to 2 across synapses)?Is there evidence supporting that?It seems to me that the calculation in the section "Analytical computation of the L2-norm and angle"--at least in principle--be extended to estimate the interference: the competition due to plasticity of neurons in the inhibitory phase increases the angle phi, and thus slows down the alignment of the weights due to plasticity of neurons in the excitatory phase.

It is a good point that in a true online scenario where newborn DGCs are born continuously, some of them would be in their early phase (excitatory phase of GABA) while others would be in their late phase (inhibitory phase of GABA). It is indeed interesting to determine if this would lead to interference in the proper maturation of any of these newborn DGCs. We tackle this question in two ways. (1) Do newborn DGCs in the late (GABA inhibitory) phase interfere with the ability of newborn DGCs in the early (GABA excitatory) phase to become selective for familiar inputs? (2) Do newborn DGCs in the early (GABA excitatory) phase interfere with the ability of newborn DGCs in the late (GABA inhibitory) phase to become selective for novel patterns?

1) In our model, newborn DGCs in the early phase of maturation (=earlyDGCs) receive indirect GABAergic excitation (through inhibitory neurons) from surrounding mature DGCs and from newborn DGCs which are in their late phase of maturation (=lateDGCs). Throughout their late phase of maturation, lateDGCs will become more and more selective for novel input patterns. Therefore, as they go through their late phase of maturation, they will push the configuration of the feedforward weight vector onto the earlyDGCs from towards the center of mass of input patterns that are well represented by mature DGCs to the center of mass of all input patterns (i.e. also the ones that were introduced during the early phase of maturation of lateDGCs). We therefore do not expect lateDGCs to interfere with the ability of earlyDGCs to become selective for familiar inputs. Rather, at the end of their maturation, earlyDGCs will either ultimately become selective for novel patterns that are similar to the ones for which mature DGCs are selective (aka different prototypes of familiar inputs), or similar to the ones for which lateDGCs eventually became selective. The alignment itself will depend on the stage of lateDGCs that indirectly activate the earlyDGCs.

2) In our model, newborn DGCs in the early phase of maturation (=earlyDGCs) do not project yet to inhibitory neurons (or any neurons). Therefore, they do not affect the circuit, and the activity of newborn DGCs in the late phase of maturation (=lateDGCs) is independent of the activity of earlyDGCs. Therefore, we do not expect a “slowing down of the alignment of the weights due to plasticity of neurons in the excitatory phase”.

Accordingly, our model does not predict that neurogenesis occurs in waves, and we are not aware of experimental evidence suggesting it.

107, Review of functional role of DGCs.Aljadeff at al., 2015,Shani-Narkiss et al., 2020suggest a dynamical role for new neurons.

Thanks, they are now included in lines 108-109.

312, It would be interesting if the advantage of adding newborn neurons stimulated with "5" to a network pretrained with "3" and "4" over a network pretrained with "3", "4", and "5" would persist if some amount of plasticity remains in mature neurons (Figure 3d).

We hope that we answered this point in the “main concern 2” and related further analyses. Plastic mature DGCs do not affect classification performance of our neurogenesis model. Rather, it does improve it in some cases. Therefore, it is closer to the main model in terms of performance, and still better than simultaneous pretraining of digits 3, 4 and 5 (our “control 1”).

614, It would be good to discuss the possibility that neurotransmitter switch (without neurogenesis) has the same functional role as GABA switch in the current model. See e.g., Li et al., (2020) J Neuroscience.Furthermore, can this model teach us anything about neurogenesis in the olfactory bulb? Is there an E to I switch there too?

We now touch upon a potential link between neurotransmitter switching and the GABA switch in our model in lines 832-837:

“Finally, while neurotransmitter switching has been observed following sustained stimulation for hours to days (Li et al., 2020), it is still unclear if it has the same functional role as the GABA switch in our model. In particular, it remains an open question if neurotransmitter switching promotes the integration of neurons in the same way as our model GABA switch does in the context of adult dentate gyrus neurogenesis.”

The suitability of our model to investigate adult olfactory bulb neurogenesis is now considered in the Discussion (lines 779-790):

“The parallel of neurogenesis in dentate gyrus and olfactory bulb suggests that similar mechanisms could be at work in both areas. Yet, even though adult olfactory bulb neurogenesis seems to have a similar functional role to adult dentate gyrus neurogenesis (Sahay et al., 2011b), follow a similar integration sequence and undergo a GABA switch from excitatory to inhibitory, the circuits are different in several aspects. First, while newborn neurons in dentate gyrus are excitatory, newborn cells in the olfactory bulb are inhibitory. Second, the newborn olfactory cells start firing action potentials only once they are well integrated (Carleton et al., 2003). Therefore, in view of a transfer of results to the olfactory bulb, it would be interesting to adjust our model of adult dentate gyrus neurogenesis accordingly. For example, a voltage-based synaptic plasticity rule could be used to account for subthreshold plasticity mechanisms (Clopath et al., 2010).”

725, Miller and Fumarola may not be the right reference to cite here. This specific nonlinearity (rectified tanh) is not standard and is not included in that paper.

Thanks for pointing out that our formulation was too dense here. In fact, there are two different aspects.

i) In Miller and Fumarola, (2012), they show the mathematical equivalence of two expressions commonly used for rate models: a “voltage equation” (their equation (1)) and a “firing rate equation” (their equation (2)). The function ‘f’ in their firing rate equation can be any nonlinear function. Therefore, our rate equation (2) corresponds to their equation (2) with a particular choice for the nonlinear function ‘f’: the rectified hyperbolic tangent.

ii) The rectified hyperbolic tangent function is a choice that “combines a hard threshold with a saturation” (equation (2.11) in the “Theoretical Neuroscience” book from Dayan and Abbott). Through rectification, negative firing rates, which are not biologically plausible, are avoided. Another way to avoid negative firing rate without rectification of the input would be to use ½ + (½)*tanh(input). The main difference is that the firing rate increases more slowly from 0 in that case, while it increases linearly with our rectified tanh. The rectified tanh is also closer to the f-I curve of spiking neuron models such as the leaky integrate-and-fire model with absolute refractory period.

To clarify these issues, in the new version of the paper we now write (lines 871-876):

“We assume that the DGCs have a frequency-current curve that is given by a rectified hyperbolic tangent (Dayan and Abbott, 2001) which is similar to the frequency-current curve of spiking neuron models with refractoriness (Gerstner et al., 2014). Moreover, we exploit the equivalence of two common firing rate equations (Miller and Fumarola, 2012) and let the firing rate nu_i_ of DGC i upon stimulation with input pattern x→ evolve according to: […]”778, Definition of quasi orthogonal is not clear. The inhibitory rates can have fluctuations and temporal dynamics of their own even if the network is assumed to be silent when each stimulus is presented. Therefore inputs might be quasi-orthogonal at one time but not at another. If in this is used just to qualitatively understand the network behavior, this somewhat sloppy definition is ok, but I think this caveat should be mentioned to avoid confusion.

Thanks for pointing this out. We now clarify that the definition is applied to the stationary state (i.e., after the transient); note that the model is noise-free and expected to be non-chaotic so that, apart from an initial transient, fluctuations cannot appear with stationary input. We now write (lines 943-954):

“We emphasize that all synaptic weights, and all presynaptic firing rates nu_j_ are non-negative: w_ij_ ≥ 0 and nu_j_ ≥ 0. […] Note that for a case without inhibitory neurons and with b_i_ -> 0, we recover the standard orthogonality condition, but for finite b_I >_ 0 quasi-orthogonality corresponds to angles larger than some reference angle.”